# Stochastic blockmodel approximation of a graphon: Theory and consistent estimation

**Edoardo M. Airoldi**
Dept. Statistics
Harvard University

**Thiago B. Costa**
SEAS, and Dept. Statistics
Harvard University

**Stanley H. Chan**
SEAS, and Dept. Statistics
Harvard University

## Abstract

Non-parametric approaches for analyzing network data based on exchangeable graph models (ExGM) have recently gained interest. The key object that defines an ExGM is often referred to as a *graphon*. This non-parametric perspective on network modeling poses challenging questions on how to make inference on the graphon underlying observed network data. In this paper, we propose a computationally efficient procedure to estimate a graphon from a set of observed networks generated from it. This procedure is based on a stochastic blockmodel approximation (SBA) of the graphon. We show that, by approximating the graphon with a stochastic block model, the graphon can be consistently estimated, that is, the estimation error vanishes as the size of the graph approaches infinity.

## 1 Introduction

Revealing hidden structures of a graph is the heart of many data analysis problems. From the well-known small-world network to the recent large-scale data collected from online service providers such as Wikipedia, Twitter and Facebook, there is always a momentum in seeking better and more informative representations of the graphs [1, 14, 29, 3, 26, 12]. In this paper, we develop a new computational tool to study one type of non-parametric representations which recently draws significant attentions from the community [4, 19, 5, 30, 23].

The root of the non-parametric model discussed in this paper is in the theory of exchangeable random arrays [2, 15, 16], and it is presented in [11] as a link connecting de Finetti's work on partial exchangeability and graph limits [20, 6]. In a nutshell, the theory predicts that every convergent sequence of graphs $(G_n)$ has a limit object that preserves many local and global properties of the graphs in the sequence. This limit object, which is called a *graphon*, can be represented by measurable functions $w : [0,1]^2 \rightarrow [0,1]$, in a way that any $w'$ obtained from measure preserving transformations of $w$ describes the same graphon.

Graphons are usually seen as kernel functions for random network models [18]. To construct an $n$-vertex random graph $\mathcal{G}(n, w)$ for a given $w$, we first assign a random label $u_i \sim \text{Uniform}[0, 1]$ to each vertex $i \in \{1, \ldots, n\}$, and connect any two vertices $i$ and $j$ with probability $w(u_i, u_j)$, *i.e.*,

$$\Pr\left(G[i, j] = 1 \mid u_i, u_j\right) = w(u_i, u_j), \qquad i, j = 1, \ldots, n, \tag{1}$$

where $G[i, j]$ denotes the $(i, j)$th entry of the adjacency matrix representing a particular realization of $\mathcal{G}(n, w)$ (See Figure 1). As an example, we note that the stochastic block-model is the case where $w(x, y)$ is a piecewise constant function.

The problem of interest is defined as follows: Given a sequence of $2T$ observed *directed* graphs $G_1, \ldots, G_{2T}$, can we make an estimate $\widehat{w}$ of $w$, such that $\widehat{w} \rightarrow w$ with high probability as $n \rightarrow \infty$? This question has been loosely attempted in the literature, but none of which has a complete solution. For example, Lloyd et al. [19] proposed a Bayesian estimator without a consistency proof; Choi and

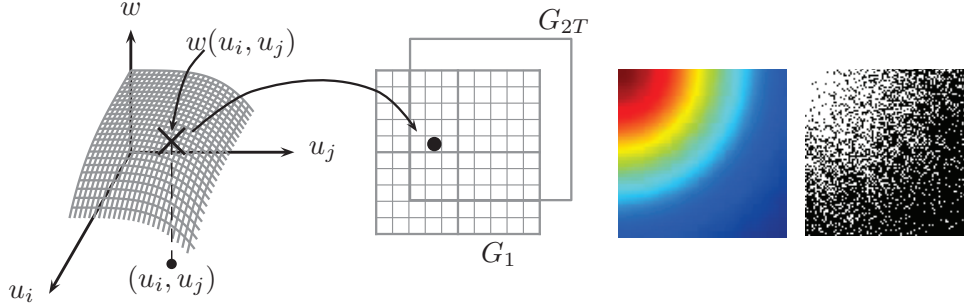

Figure 1: [Left] Given a graphon $w : [0,1]^2 \to [0,1]$, we draw i.i.d. samples $u_i$, $u_j$ from Uniform[0,1] and assign $G_t[i,j] = 1$ with probability $w(u_i, u_j)$, for $t = 1, \ldots, 2T$. [Middle] Heat map of a graphon $w$. [Right] A random graph generated by the graphon shown in the middle. Rows and columns of the graph are ordered by increasing $u_i$, instead of $i$ for better visualization.

Wolfe [9] studied the consistency properties, but did not provide algorithms to estimate the graphon. To the best of our knowledge, the only method that estimates graphons consistently, besides ours, is USVT [8]. However, our algorithm has better complexity and outperforms USVT in our simulations. More recently, other groups have begun exploring approaches related to ours [28, 24].

The proposed approximation procedure requires $w$ to be piecewise Lipschitz. The basic idea is to approximate $w$ by a two-dimensional step function $\widehat{w}$ with diminishing intervals as $n$ increases. The proposed method is called the stochastic blockmodel approximation (SBA) algorithm, as the idea of using a two-dimensional step function for approximation is equivalent to using the stochastic block models [10, 22, 13, 7, 25]. The SBA algorithm is defined up to permutations of the nodes, so the estimated graphon is *not* canonical. However, this does not affect the consistency properties of the SBA algorithm, as the consistency is measured w.r.t. the graphon that generates the graphs.

## 2 Stochastic blockmodel approximation: Procedure

In this section we present the proposed SBA algorithm and discuss its basic properties.

### 2.1 Assumptions on graphons

We assume that $w$ is *piecewise Lipschitz, i.e.,* there exists a sequence of non-overlaping intervals $I_k = [\alpha_{k-1}, \alpha_k]$ defined by $0 = \alpha_0 < \ldots < \alpha_K = 1$, and a constant $L > 0$ such that, for any $(x_1, y_1)$ and $(x_2, y_2) \in I_{ij} = I_i \times I_j$,

$$|w(x_1, y_1) - w(x_2, y_2)| \leq L \left( |x_1 - x_2| + |y_1 - y_2| \right).$$

For generality we assume $w$ to be asymmetric *i.e.,* $w(u, v) \neq w(v, u)$, so that symmetric graphons can be considered as a special case. Consequently, a random graph $\mathcal{G}(n, w)$ generated by $w$ is directed, *i.e.,* $G[i, j] \neq G[j, i]$.

### 2.2 Similarity of graphon slices

The intuition of the proposed SBA algorithm is that if the graphon is smooth, neighboring cross-sections of the graphon should be similar. In other words, if two labels $u_i$ and $u_j$ are close *i.e.,* $|u_i - u_j| \approx 0$, then the difference between the row slices $|w(u_i, \cdot) - w(u_j, \cdot)|$ and the column slices $|w(\cdot, u_i) - w(\cdot, u_j)|$ should also be small. To measure the similarity between two labels using the graphon slices, we define the following distance

$$d_{ij} = \frac{1}{2} \left( \int_0^1 [w(x, u_i) - w(x, u_j)]^2 \, dx + \int_0^1 [w(u_i, y) - w(u_j, y)]^2 \, dy \right). \qquad (2)$$

Thus, $d_{ij}$ is small only if both row and column slices of the graphon are similar.

The usage of $d_{ij}$ for graphon estimation will be discussed in the next subsection. But before we proceed, it should be noted that in practice $d_{ij}$ has to be estimated from the observed graphs $G_1, \ldots, G_{2T}$. To derive an estimator $\widehat{d}_{ij}$ of $d_{ij}$, it is helpful to express $d_{ij}$ in a way that the estimators can be easily obtained. To this end, we let

$$c_{ij} = \int_0^1 w(x, u_i)w(x, u_j)dx \qquad \text{and} \qquad r_{ij} = \int_0^1 w(u_i, y)w(u_j, y)dy,$$

and express $d_{ij}$ as $d_{ij} = \frac{1}{2}\Big[(c_{ii} - c_{ij} - c_{ji} + c_{jj}) + (r_{ii} - r_{ij} - r_{ji} + r_{jj})\Big]$. Inspecting this expression, we consider the following estimators for $c_{ij}$ and $r_{ij}$:

$$\widehat{c}_{ij}^k = \frac{1}{T^2}\left(\sum_{1 \le t_1 \le T} G_{t_1}[k, i]\right)\left(\sum_{T < t_2 \le 2T} G_{t_2}[k, j]\right), \tag{3}$$

$$\widehat{r}_{ij}^k = \frac{1}{T^2}\left(\sum_{1 \le t_1 \le T} G_{t_1}[i, k]\right)\left(\sum_{T < t_2 \le 2T} G_{t_2}[j, k]\right). \tag{4}$$

Here, the superscript $k$ can be interpreted as the dummy variables $x$ and $y$ in defining $c_{ij}$ and $r_{ij}$, respectively. Summing all possible $k$'s yields an estimator $\widehat{d}_{ij}$ that looks similar to $d_{ij}$:

$$\widehat{d}_{ij} = \frac{1}{2}\left[\frac{1}{S}\sum_{k \in \mathcal{S}}\left\{\left(\widehat{r}_{ii}^k - \widehat{r}_{ij}^k - \widehat{r}_{ji}^k + \widehat{r}_{jj}^k\right) + \left(\widehat{c}_{ii}^k - \widehat{c}_{ij}^k - \widehat{c}_{ji}^k + \widehat{c}_{jj}^k\right)\right\}\right], \tag{5}$$

where $\mathcal{S} = \{1, \ldots, n\}\backslash\{i, j\}$ is the set of summation indices.

The motivation of defining the estimators in (3) and (4) is that a row of the adjacency matrix $G[i, \cdot]$ is fully characterized by the corresponding row of the graphon $w(u_i, \cdot)$. Thus the expected value of $\frac{1}{T}\left(\sum_{1 \le t_1 \le T} G_{t_1}[i, \cdot]\right)$ is $w(u_i, \cdot)$, and hence $\frac{1}{S}\sum_{k \in \mathcal{S}} \widehat{r}_{ij}^k$ is an estimator for $r_{ij}$. To theoretically justify this intuition, we will show in Section 3 that $\widehat{d}_{ij}$ is indeed a good estimator: it is not only unbiased, but is also concentrated round $d_{ij}$ for large $n$. Furthermore, we will show that it is possible to use a random subset of $\mathcal{S}$ instead of $\{1, \ldots, n\}\backslash\{i, j\}$ to achieve the same asymptotic behavior. As a result, the estimation of $d_{ij}$ can be performed locally in a neighborhood of $i$ and $j$, instead of all $n$ vertices.

## 2.3 Blocking the vertices

The similarity metric $\widehat{d}_{ij}$ discussed above suggests one simple method to approximate $w$ by a piece-wise constant function $\widehat{w}$ (*i.e.*, a stochastic block-model). Given $G_1, \ldots, G_{2T}$, we can cluster the (unknown) labels $\{u_1, \ldots, u_n\}$ into $K$ blocks $\widehat{B}_1, \ldots, \widehat{B}_K$ using a procedure described below. Once the blocks $\widehat{B}_1, \ldots, \widehat{B}_K$ are defined, we can then determine $\widehat{w}(u_i, u_j)$ by computing the empirical frequency of edges that are present across blocks $\widehat{B}_i$ and $\widehat{B}_j$:

$$\widehat{w}(u_i, u_j) = \frac{1}{|\widehat{B}_i|\,|\widehat{B}_j|}\sum_{i_x \in \widehat{B}_i}\sum_{j_y \in \widehat{B}_j}\frac{1}{2T}\left(G_1[i_x, j_y] + G_2[i_x, j_y] + \ldots + G_{2T}[i_x, j_y]\right), \tag{6}$$

where $\widehat{B}_i$ is the block containing $u_i$ so that summing $G_t[x, y]$ over $x \in \widehat{B}_i$ and $y \in \widehat{B}_j$ yields an estimate of the expected number of edges linking block $\widehat{B}_i$ and $\widehat{B}_j$.

To cluster the unknown labels $\{u_1, \ldots, u_n\}$ we propose a greedy approach as shown in Algorithm 1. Starting with $\Omega = \{u_1, \ldots, u_n\}$, we randomly pick a node $i_p$ and call it the *pivot*. Then for all other vertices $i_v \in \Omega\backslash\{i_p\}$, we compute the distance $\widehat{d}_{i_p, i_v}$ and check whether $\widehat{d}_{i_p, i_v} < \Delta^2$ for some precision parameter $\Delta > 0$. If $\widehat{d}_{i_p, i_v} < \Delta^2$, then we assign $i_v$ to the same block as $i_p$. Therefore, after scanning through $\Omega$ once, a block $\widehat{B}_1 = \{i_p, i_{v_1}, i_{v_2}, \ldots\}$ will be defined. By updating $\Omega$ as $\Omega \leftarrow \Omega\backslash\widehat{B}_1$, the process repeats until $\Omega = \emptyset$.

The proposed greedy algorithm is only a local solution in a sense that it does not return the globally optimal clusters. However, as will be shown in Section 3, although the clustering algorithm is not globally optimal, the estimated graphon $\widehat{w}$ is still guaranteed to be a consistent estimate of the true graphon $w$ as $n \to \infty$. Since the greedy algorithm is numerically efficient, it serves as a practical computational tool to estimate $w$.

## 2.4 Main algorithm

---
**Algorithm 1** Stochastic blockmodel approximation
---
Input: A set of observed graphs $G_1, \ldots, G_{2T}$ and the precision parameter $\Delta$.
Output: Estimated stochastic blocks $\widehat{B}_1, \ldots, \widehat{B}_K$.
Initialize: $\Omega = \{1, \ldots, n\}$, and $k = 1$.
**while** $\Omega \neq \emptyset$ **do**
    Randomly choose a vertex $i_p$ from $\Omega$ and assign it as the pivot for $\widehat{B}_k$: $\widehat{B}_k \leftarrow i_p$.
    **for** Every other vertices $i_v \in \Omega \backslash \{i_p\}$ **do**
        Compute the distance estimate $\widehat{d}_{i_p, i_v}$.
        If $\widehat{d}_{i_p, i_v} \leq \Delta^2$, then assign $i_v$ as a member of $\widehat{B}_k$: $\widehat{B}_k \leftarrow i_v$.
    **end for**
    Update $\Omega$: $\Omega \leftarrow \Omega \backslash \widehat{B}_k$.
    Update counter: $k \leftarrow k + 1$.
**end while**
---

Algorithm 1 illustrates the pseudo-code for the proposed stochastic block-model approximation. The complexity of this algorithm is $\mathcal{O}(TSKn)$, where $T$ is half the number of observations, $S$ is the size of the neighborhood, $K$ is the number of blocks and $n$ is number of vertices of the graph.

## 3 Stochastic blockmodel approximation: Theory of estimation

In this section we present the theoretical aspects of the proposed SBA algorithm. We will first discuss the properties of the estimator $\widehat{d}_{ij}$, and then show the consistency of the estimated graphon $\widehat{w}$. Details of the proofs can be found in the supplementary material.

### 3.1 Concentration analysis of $\widehat{d}_{ij}$

Our first theorem below shows that the proposed estimator $\widehat{d}_{ij}$ is both unbiased, and is concentrated around its expected value $d_{ij}$.

**Theorem 1.** *The estimator $\widehat{d}_{ij}$ for $d_{ij}$ is unbiased, i.e., $\mathbb{E}[\widehat{d}_{ij}] = d_{ij}$. Further, for any $\epsilon > 0$,*

$$\Pr\left[\left|\widehat{d}_{ij} - d_{ij}\right| > \epsilon\right] \leq 8e^{-\frac{S\epsilon^2}{32/T + 8\epsilon/3}}, \tag{7}$$

*where $S$ is the size of the neighborhood $\mathcal{S}$, and $2T$ is the number of observations.*

*Proof.* Here we only highlight the important steps to present the intuition. The basic idea of the proof is to zoom-in a microscopic term of $\widehat{r}_{ij}^k$ and show that it is unbiased. To this end, we use the fact that $G_{t_1}[i, k]$ and $G_{t_2}[j, k]$ are conditionally independent on $u_k$ to show

$$\begin{aligned}
\mathbb{E}[G_{t_1}[i, k] G_{t_2}[j, k] \mid u_k] &= \Pr[G_{t_1}[i, k] = 1, G_{t_2}[j, k] = 1 \mid u_k] \\
&\overset{(a)}{=} \Pr[G_{t_1}[i, k] = 1 \mid u_k] \Pr[G_{t_2}[j, k] = 1 \mid u_k] \\
&= w(u_i, u_k) w(u_j, u_k),
\end{aligned}$$

which then implies $\mathbb{E}[\widehat{r}_{ij}^k \mid u_k] = w(u_i, u_k) w(u_j, u_k)$, and by iterated expectation we have $\mathbb{E}[\widehat{r}_{ij}^k] = \mathbb{E}[\mathbb{E}[\widehat{r}_{ij}^k \mid u_k]] = r_{ij}$. The concentration inequality follows from a similar idea to bound the variance of $\widehat{r}_{ij}^k$ and apply Bernstein's inequality. $\qquad\square$

That $G_{t_1}[i,k]$ and $G_{t_2}[j,k]$ are conditionally independent on $u_k$ is a critical fact for the success of the proposed algorithm. It also explains why at least 2 independently observed graphs are necessary, for otherwise we cannot separate the probability in the second equality above marked with $(a)$.

## 3.2 Choosing the number of blocks

The performance of the Algorithm 1 is sensitive to the number of blocks it defines. On the one hand, it is desirable to have more blocks so that the graphon can be finely approximated. But on the other hand, if the number of blocks is too large then each block will contain only few vertices. This is bad because in order to estimate the value on each block, a sufficient number of vertices in each block is required. The trade-off between these two cases is controlled by the precision parameter $\Delta$: a large $\Delta$ generates few large clusters, while small $\Delta$ generates many small clusters. A precise relationship between the $\Delta$ and $K$, the number of blocks generated the algorithm, is given in Theorem 2.

**Theorem 2.** *Let $\Delta$ be the accuracy parameter and $K$ be the number of blocks estimated by Algorithm 1, then*

$$\Pr\left[K > \frac{QL\sqrt{2}}{\Delta}\right] \le 8n^2 e^{-\frac{S\Delta^4}{128/T+16\Delta^2/3}}, \tag{8}$$

*where $L$ is the Lipschitz constant and $Q$ is the number of Lipschitz blocks in $w$.*

In practice, we estimate $\Delta$ using a cross-validation scheme to find the optimal 2D histogram bin width [27]. The idea is to test a sequence of potential values of $\Delta$ and seek the one that minimizes the cross validation risk, defined as

$$\widehat{J}(\Delta) = \frac{2}{h(n-1)} - \frac{n+1}{h(n-1)} \sum_{j=1}^{K} \widehat{p}_j^2, \tag{9}$$

where $\widehat{p}_j = |\widehat{B}_j|/n$ and $h = 1/K$. Algorithm 2 details the proposed cross-validation scheme.

---

**Algorithm 2** Cross Validation

Input: Graphs $G_1, \ldots, G_{2T}$.
Output: Blocks $\widehat{B}_1, \ldots, \widehat{B}_K$, and optimal $\Delta$.
**for** a sequence of $\Delta$'s **do**
    Estimate blocks $\widehat{B}_1, \ldots, \widehat{B}_K$ from $G_1, \ldots, G_{2T}$. [Algorithm 1]
    Compute $\widehat{p}_j = |\widehat{B}_j|/n$, for $j = 1, \ldots, K$.
    Compute $\widehat{J}(\Delta) = \frac{2}{h(n-1)} - \frac{n+1}{h(n-1)} \sum_{j=1}^{K} \widehat{p}_j^2$, with $h = 1/K$.
**end for**
Pick the $\Delta$ with minimum $\widehat{J}(\Delta)$, and the corresponding $\widehat{B}_1, \ldots, \widehat{B}_K$.

---

## 3.3 Consistency of $\widehat{w}$

The goal of our next theorem is to show that $\widehat{w}$ is a consistent estimate of $w$, *i.e.*, $\widehat{w} \to w$ as $n \to \infty$. To begin with, let us first recall two commonly used metric:

**Definition 1.** *The mean squared error (MSE) and mean absolute error (MAE) are defined as*

$$\text{MSE}(\widehat{w}) = \frac{1}{n^2} \sum_{i_v=1}^{n} \sum_{j_v=1}^{n} \left(w(u_{i_v}, u_{j_v}) - \widehat{w}(u_{i_v}, u_{j_v})\right)^2$$

$$\text{MAE}(\widehat{w}) = \frac{1}{n^2} \sum_{i_v=1}^{n} \sum_{j_v=1}^{n} \left|w(u_{i_v}, u_{j_v}) - \widehat{w}(u_{i_v}, u_{j_v})\right|.$$

**Theorem 3.** *If $S \in \Theta(n)$ and $\Delta \in \omega\left(\left(\frac{\log(n)}{n}\right)^{\frac{1}{4}}\right) \cap o(1)$, then*

$$\lim_{n\to\infty} \mathbb{E}[\text{MAE}(\widehat{w})] = 0 \qquad \text{and} \qquad \lim_{n\to\infty} \mathbb{E}[\text{MSE}(\widehat{w})] = 0.$$

*Proof.* The details of the proof can be found in the supplementary material . Here we only outline the key steps to present the intuition of the theorem. The goal of Theorem 3 is to show convergence of $|\widehat{w}(u_i, u_j) - w(u_i, u_j)|$. The idea is to consider the following two quantities:

$$\overline{w}(u_i, u_j) = \frac{1}{|\widehat{B}_i||\widehat{B}_j|} \sum_{i_x \in \widehat{B}_i} \sum_{j_x \in \widehat{B}_j} w(u_{i_x}, u_{j_x}),$$

$$\widehat{w}(u_i, u_j) = \frac{1}{|\widehat{B}_i||\widehat{B}_j|} \sum_{i_x \in \widehat{B}_i} \sum_{j_y \in \widehat{B}_j} \frac{1}{2T} \left(G_1[i_x, j_y] + G_2[i_x, j_y] + \ldots + G_{2T}[i_x, j_y]\right),$$

so that if we can bound $|\overline{w}(u_i, u_j) - w(u_i, u_j)|$ and $|\overline{w}(u_i, u_j) - \widehat{w}(u_i, u_j)|$, then consequently $|\widehat{w}(u_i, u_j) - w(u_i, u_j)|$ can also be bounded.

The bound for the first term $|\overline{w}(u_i, u_j) - w(u_i, u_j)|$ is shown in Lemma 1: By Algorithm 1, any vertex $i_v \in \widehat{B}_i$ is guaranteed to be within a distance $\Delta$ from the pivot of $\widehat{B}_i$. Since $\overline{w}(u_i, u_j)$ is an average over $\widehat{B}_i$ and $\widehat{B}_j$, by Theorem 1 a probability bound involving $\Delta$ can be obtained.

The bound for the second term $|\overline{w}(u_i, u_j) - \widehat{w}(u_i, u_j)|$ is shown in Lemma 2. Different from Lemma 1, here we need to consider two possible situations: either the intermediate estimate $\overline{w}(u_i, u_j)$ is close to the ground truth $w(u_i, u_j)$, or $\overline{w}(u_i, u_j)$ is far from the ground truth $w(u_i, u_j)$. This accounts for the sum in Lemma 2. Individual bounds are derived based on Lemma 1 and Theorem 1.

Combining Lemma 1 and Lemma 2, we can then bound the error and show convergence. □

**Lemma 1.** *For any* $i_v \in \widehat{B}_i$ *and* $j_v \in \widehat{B}_j$,

$$\Pr\left[|\overline{w}(u_i, u_j) - w(u_{i_v}, u_{j_v})| > 8\Delta^{1/2}L^{1/4}\right] \leq 32|\widehat{B}_i||\widehat{B}_j|e^{-\frac{S\Delta^4}{32/T + 8\Delta^2/3}}. \tag{10}$$

**Lemma 2.** *For any* $i_v \in \widehat{B}_i$ *and* $j_v \in \widehat{B}_j$,

$$\Pr\left[|\widehat{w}_{ij} - \overline{w}_{ij}| > 8\Delta^{1/2}L^{1/4}\right] \leq 2e^{-256(T|\widehat{B}_i||\widehat{B}_j|\sqrt{L}\Delta)} + 32|\widehat{B}_i|^2|\widehat{B}_j|^2 e^{-\frac{S\Delta^4}{32/T + 8\Delta^2/3}}. \tag{11}$$

The condition $S \in \Theta(n)$ is necessary to make Theorem 3 valid, because if $S$ is independent of $n$, the right hand sides of (10) and (11) cannot approach 0 even if $n \to \infty$. The condition on $\Delta$ is also important as it forces the numerators and denominators in the exponentials of (10) and (11) to be well behaved.

## 4 Experiments

In this section we evaluate the proposed SBA algorithm by showing some empirical results. For the purpose of comparison, we consider (i) the universal singular value thresholding (USVT) [8]; (ii) the largest-gap algorithm (LG) [7]; (iii) matrix completion from few entries (OptSpace) [17].

### 4.1 Estimating stochastic blockmodels

**Accuracy as a function of growing graph size.** Our first experiment is to evaluate the proposed SBA algorithm for estimating stochastic blockmodels. For this purpose, we generate (arbitrarily) a graphon

$$w = \begin{bmatrix} 0.8 & 0.9 & 0.4 & 0.5 \\ 0.1 & 0.6 & 0.3 & 0.2 \\ 0.3 & 0.2 & 0.8 & 0.3 \\ 0.4 & 0.1 & 0.2 & 0.9 \end{bmatrix}, \tag{12}$$

which represents a piecewise constant function with $4 \times 4$ equi-space blocks.

Since USVT and LG use only one observed graph whereas the proposed SBA require at least 2 observations, in order to make the comparison fair, we use half of the nodes for SBA by generating two independent $\frac{n}{2} \times \frac{n}{2}$ observed graphs. For USVT and LG, we use one $n \times n$ observed graph.

Figure 2(a) shows the asymptotic behavior of the algorithms when $n$ grows. Figure 2(b) shows the estimation error of SBA algorithm as $T$ grows for graphs of size 200 vertices.

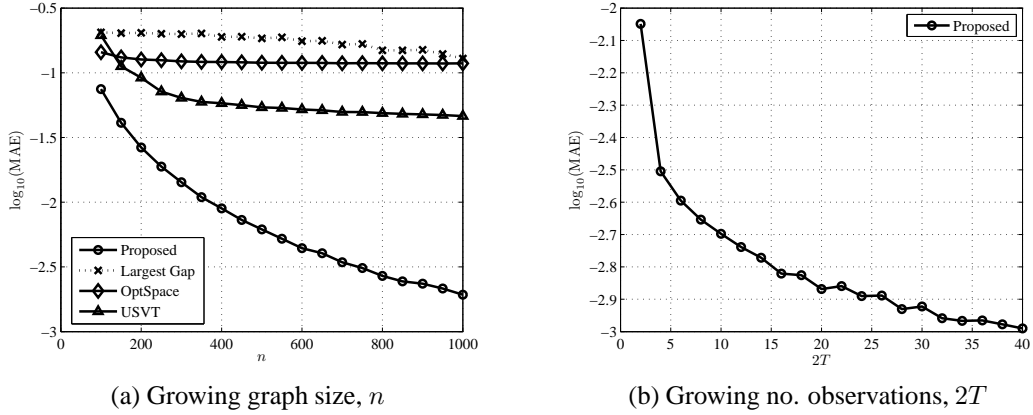

(a) Growing graph size, $n$       (b) Growing no. observations, $2T$

Figure 2: (a) MAE reduces as graph size grows. For the fairness of the amount of data that can be used, we use $\frac{n}{2} \times \frac{n}{2} \times 2$ observations for SBA, and $n \times n \times 1$ observation for USVT [8] and LG [7]. (b) MAE of the proposed SBA algorithm reduces when more observations $T$ is available. Both plots are averaged over 100 independent trials.

**Accuracy as a function of growing number of blocks.** Our second experiment is to evaluate the performance of the algorithms as $K$, the number of blocks, increases. To this end, we consider a sequence of $K$, and for each $K$ we generate a graphon $w$ of $K \times K$ blocks. Each entry of the block is a random number generated from Uniform$[0, 1]$. Same as the previous experiment, we fix $n = 200$ and $T = 1$. The experiment is repeated over 100 trials so that in every trial a different graphon is generated. The result shown in Figure 3(a) indicates that while estimation error increases as $K$ grows, the proposed SBA algorithm still attains the lowest MAE for all $K$.

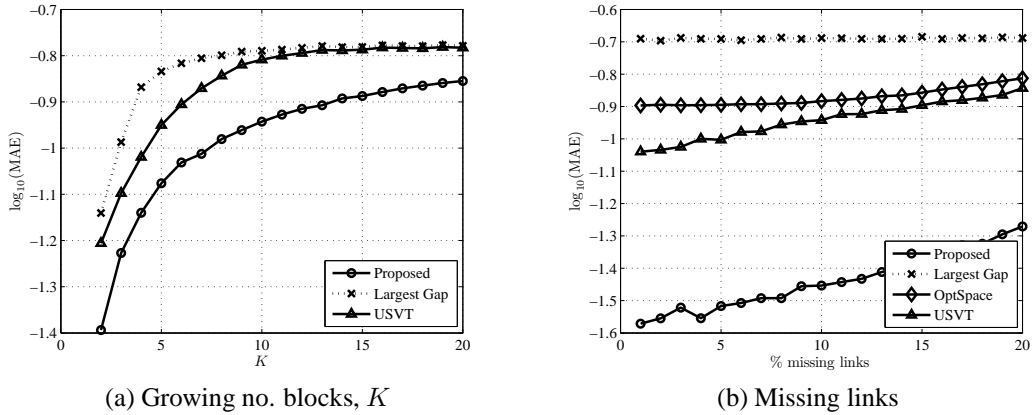

(a) Growing no. blocks, $K$       (b) Missing links

Figure 3: (a) As $K$ increases, MAE of all three algorithm increases but SBA still attains the lowest MAE. Here, we use $\frac{n}{2} \times \frac{n}{2} \times 2$ observations for SBA, and $n \times n \times 1$ observation for USVT [8] and LG [7]. (b) Estimation of graphon in the presence of missing links: As the amount of missing links increases, estimation error also increases.

## 4.2 Estimation with missing edges

Our next experiment is to evaluate the performance of proposed SBA algorithm when there are missing edges in the observed graph. To model missing edges, we construct an $n \times n$ binary matrix $M$ with probability $\Pr[M[i, j] = 0] = \xi$, where $0 \le \xi \le 1$ defines the percentage of missing edges. Given $\xi$, $2T$ matrices are generated with missing edges, and the observed graphs are defined as $M_1 \odot G_1, \ldots, M_{2T} \odot G_{2T}$, where $\odot$ denotes the element-wise multiplication. The goal is to study how well SBA can reconstruct the graphon $\widehat{w}$ in the presence of missing links.

The modification of the proposed SBA algorithm for the case missing links is minimal: when computing (6), instead of averaging over all $i_x \in \widehat{B}_i$ and $j_y \in \widehat{B}_j$, we only average $i_x \in \widehat{B}_i$ and $j_y \in \widehat{B}_j$ that are not masked out by all $M$'s. Figure 3(b) shows the result of average over 100 independent trials. Here, we consider the graphon given in (12), with $n = 200$ and $T = 1$. It is evident that SBA outperforms its counterparts at a lower rate of missing links.

## 4.3 Estimating continuous graphons

Our final experiment is to evaluate the proposed SBA algorithm in estimating continuous graphons. Here, we consider two of the graphons reported in [8]:

$$w_1(u,v) = \frac{1}{1 + \exp\{-50(u^2 + v^2)\}}, \quad \text{and} \quad w_2(u,v) = uv,$$

where $u, v \in [0, 1]$. Here, $w_2$ can be considered as a special case of the Eigenmodel [13] or latent feature relational model [21].

The results in Figure 4 shows that while both algorithms have improved estimates when $n$ grows, the performance depends on which of $w_1$ and $w_2$ that we are studying. This suggests that in practice the choice of the algorithm should depend on the expected structure of the graphon to be estimated: If the graph generated by the graphon demonstrates some low-rank properties, then USVT is likely to be a better option. For more structured or complex graphons the proposed procedure is recommended.

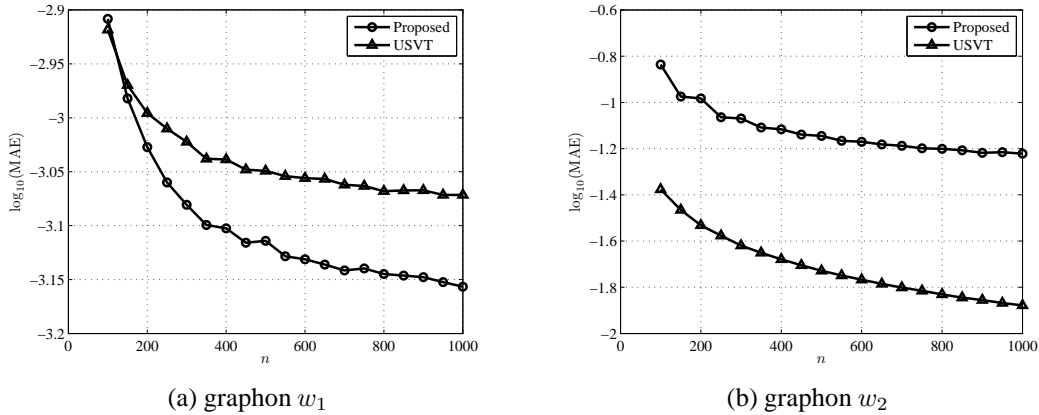

(a) graphon $w_1$                        (b) graphon $w_2$

Figure 4: Comparison between SBA and USVT in estimating two continuous graphons $w_1$ and $w_2$. Evidently, SBA performs better for $w_1$ (high-rank) and worse for $w_2$ (low-rank).

## 5 Concluding remarks

We presented a new computational tool for estimating graphons. The proposed algorithm approximates the continuous graphon by a stochastic block-model, in which the first step is to cluster the unknown vertex labels into blocks by using an empirical estimate of the distance between two graphon slices, and the second step is to build an empirical histogram to estimate the graphon. Complete consistency analysis of the algorithm is derived. The algorithm was evaluated experimentally, and we found that the algorithm is effective in estimating block structured graphons.

Implementation of the SBA algorithm is available online at https://github.com/airoldilab/SBA.

**Acknowledgments**. EMA is partially supported by NSF CAREER award IIS-1149662, ARO MURI award W911NF-11-1-0036, and an Alfred P. Sloan Research Fellowship. SHC is partially supported by a Croucher Foundation Post-Doctoral Research Fellowship.

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
