[Supplementary Material · supp_v6.pdf]

# Supplementary Material

## 1 Proofs for Section 3.1

**Theorem 1.** *The estimator $\widehat{d}_{ij}$ for $d_{ij}$ is unbiased. Further, for any $\epsilon > 0$, if the graph is directed, then*

$$\Pr\left[\left|\widehat{d}_{ij} - d_{ij}\right| > \epsilon\right] \leq 8e^{-\frac{S\epsilon^2}{32/T+8\epsilon/3}}, \tag{1}$$

*and if the graph is un-directed, then*

$$\Pr\left[\left|\widehat{d}_{ij} - d_{ij}\right| > \epsilon\right] \leq 8e^{-\frac{S\epsilon^2}{64/T+8\epsilon/3}}, \tag{2}$$

*where $S$ is the size of the sampling neighborhood $\mathcal{S}$, and $2T$ is the number of observations.*

*Proof.* First, for given $u_i$ and $u_j$, let us define the following two quantities

$$c_{ij} \overset{\text{def}}{=} \int_0^1 w(x, u_i)w(x, u_j)dx,$$

$$r_{ij} \overset{\text{def}}{=} \int_0^1 w(u_i, y)w(u_j, y)dy.$$

Consequently, we express $d_{ij}$ as

$$d_{ij} \overset{\text{def}}{=} \frac{1}{2}\left(\int_0^1 (w(u_i, y) - w(u_j, y))^2 dy + \int_0^1 (w(x, u_i) - w(x, u_j))^2 dx\right)$$

$$= \frac{1}{2}\left[(r_{ii} - r_{ij} - r_{ji} + r_{jj}) + (c_{ii} - c_{ij} - c_{ji} + c_{jj})\right].$$

In order to study $\widehat{d}_{ij}$ (the estimator of $d_{ij}$), it is desired to express $\widehat{d}_{ij}$ in the same form of $d_{ij}$:

$$\widehat{d}_{ij} = \frac{1}{S}\sum_{k\in\mathcal{S}}\left\{\frac{1}{2}\left[\left(\widehat{r}_{ii}^k - \widehat{r}_{ij}^k - \widehat{r}_{ji}^k + \widehat{r}_{jj}^k\right) + \left(\widehat{c}_{ii}^k - \widehat{c}_{ij}^k - \widehat{c}_{ji}^k + \widehat{c}_{jj}^k\right)\right]\right\}, \tag{3}$$

where $\mathcal{S} = \{1, \ldots, n\}\backslash\{i, j\}$ is the sampling neighborhood, and $S = |\mathcal{S}|$. In (3), individual components are defined as

$$\widehat{c}_{ij}^k = \frac{1}{T^2}\left(\sum_{1\leq t_1\leq T} G_{t_1}[k, i]\right)\left(\sum_{T<t_2\leq 2T} G_{t_2}[k, j]\right),$$

$$\widehat{r}_{ij}^k = \frac{1}{T^2}\left(\sum_{1\leq t_1\leq T} G_{t_1}[i, k]\right)\left(\sum_{T<t_2\leq 2T} G_{t_2}[j, k]\right).$$

Thus, if we can show that $\widehat{r}_{ij}^k$ and $\widehat{c}_{ij}^k$ are unbiased estimators of $r_{ij}$ and $c_{ij}$, i.e., $\mathbb{E}[\widehat{r}_{ij}^k] = r_{ij}$ and $\mathbb{E}[\widehat{c}_{ij}^k] = c_{ij}$, then by linearity of expectation, $\widehat{d}_{ij}$ will be an unbiased estimator of $d_{ij}$.

To this end, we consider the conditional expectation of $G_{t_1}[i,k]G_{t_2}[j,k]$ given $u_k$:

$$\mathbb{E}[G_{t_1}[i,k]G_{t_2}[j,k] \mid u_k] = 1 \cdot \Pr\left[G_{t_1}[i,k]G_{t_2}[j,k] = 1 \;\Big|\; u_k\right] + 0 \cdot \Pr\left[G_{t_1}[i,k]G_{t_2}[j,k] = 1 \;\Big|\; u_k\right]$$

$$= \Pr\left[G_{t_1}[i,k] = 1 \text{ and } G_{t_2}[j,k] = 1 \;\Big|\; u_k\right]$$

$$= \Pr[G_{t_1}[i,k] = 1 \mid u_k] \cdot \Pr[G_{t_2}[j,k] = 1 \mid u_k], \quad \text{because } G_{t_1}[i,k] \perp G_{t_2}[j,k]$$

$$= w(u_i, u_k)w(u_j, u_k). \tag{4}$$

Therefore,

$$\mathbb{E}\left[\widehat{r}_{ij}^{k} \mid u_k\right] = \frac{1}{T^2}\left(\sum_{t_2=T+1}^{2T}\sum_{t_1=1}^{T}\mathbb{E}\left[G_{t_1}[i,k]G_{t_2}[j,k] \mid u_k\right]\right)$$

$$= \frac{1}{T^2}\left(\sum_{t_2=T+1}^{2T}\sum_{t_1=1}^{T}w(u_i, u_k)w(u_j, u_k)\right), \quad \text{by substituting (4)}$$

$$= w(u_i, u_k)w(u_j, u_k). \tag{5}$$

Then, by the law of iterated expectations, we have

$$\mathbb{E}\left[\widehat{r}_{ij}^{k}\right] = \mathbb{E}\left[\mathbb{E}\left[\widehat{r}_{ij}^{k} \mid u_k\right]\right]$$

$$= \mathbb{E}\left[w(u_i, u_k)w(u_j, u_k)\right], \quad \text{by substituting (5)}$$

$$= \int_0^1 w(u_i, v)w(u_j, v)dv, \quad \text{because } u_k \sim \text{Uniform}(0,1)$$

$$= r_{ij}. \tag{6}$$

Therefore, $\widehat{r}_{ij}^{k}$ is an unbiased estimator of $r_{ij}$. The proof of $\widehat{c}_{ij}$ can be similarly proved by switching roles of $G_t[i,k]$ to $G_t[k,i]$. Since $\widehat{r}_{ij}^{k}$ and $\widehat{c}_{ij}^{k}$ are both unbiased, $\widehat{d}_{ij}$ must be unbiased.

Now we proceed to prove the second part of the theorem. We first claim that

$$\text{Var}\left[\widehat{r}_{ij}^{k}\right] \le 2/T \quad \text{and} \quad \text{Var}\left[\widehat{c}_{ij}^{k}\right] \le 2/T. \tag{7}$$

To prove this, we note that

$$\text{Var}\left[\widehat{r}_{ij}^{k}\right] = \text{Var}\left[\sum_{t_2=T+1}^{2T}\sum_{t_1=1}^{T}G_{t_1}[ik]G_{t_2}[jk]\right]$$

$$= \sum_{t_2=T+1}^{2T}\sum_{t_1=1}^{T}\text{Var}\left[G_{t_1}[ik]G_{t_2}[jk]\right]$$

$$+ \sum_{\substack{\tau_2=T+1 \\ \tau_2 \ne t_2}}^{2T}\sum_{t_2=T+1}^{2T}\sum_{\substack{\tau_1=1 \\ \tau_1 \ne t_1}}^{T}\sum_{t_1=1}^{T}\text{Cov}\left[G_{t_1}[ik]G_{t_2}[jk],\ G_{\tau_1}[ik]G_{\tau_2}[jk]\right]$$

We consider three cases:

**Case 1**. First assume $\tau_1 \neq t_1$ and $\tau_2 \neq t_2$. (Occurs $(T-1)^2 T^2$ times.)

$$
\begin{aligned}
&\text{Cov}\Big[ G_{t_1}[ik]G_{t_2}[jk],\ G_{\tau_1}[ik]G_{\tau_2}[jk] \Big] \\
&= \mathbb{E}\Big[ \big(G_{t_1}[ik]G_{t_2}[jk] - \mathbb{E}[G_{t_1}[ik]G_{t_2}[jk]]\big)\big(G_{\tau_1}[ik]G_{\tau_2}[jk] - \mathbb{E}[G_{\tau_1}[ik]G_{\tau_2}[jk]]\big) \Big] \\
&= \mathbb{E}\Big[ \big(G_{t_1}[ik]G_{t_2}[jk] - w_{ik}w_{jk}\big)\big(G_{\tau_1}[ik]G_{\tau_2}[jk] - w_{ik}w_{jk}\big) \Big] \\
&= \mathbb{E}\Big[ G_{t_1}[ik]G_{t_2}[jk]G_{\tau_1}[ik]G_{\tau_2}[jk] \Big] - \mathbb{E}\Big[ G_{\tau_1}[ik]G_{\tau_2}[jk] \Big] w_{ik}w_{jk} - \mathbb{E}\Big[ G_{t_1}[ik]G_{t_2}[jk] \Big] w_{ik}w_{jk} + w_{ik}^2 w_{jk}^2 \\
&= \mathbb{E}\Big[ G_{t_1}[ik]G_{t_2}[jk]G_{\tau_1}[ik]G_{\tau_2}[jk] \Big] - w_{ik}^2 w_{jk}^2 \qquad\qquad\qquad\qquad\qquad\qquad (8)
\end{aligned}
$$

The first term in (8) is $\mathbb{E}\Big[ G_{t_1}[ik]G_{t_2}[jk]G_{\tau_1}[ik]G_{\tau_2}[jk] \Big] = w_{ik}^2 w_{jk}^2$ because $G_{t_1}[ik]$, $G_{t_2}[jk]$, $G_{\tau_1}[ik]$ and $G_{\tau_2}[jk]$ are all independent. Therefore, the overall sum in (8) is 0.

**Case 2**. Next assume that $\tau_1 \neq t_1$ but $\tau_2 = t_2$. (Occurs $(T-1)T^2$ times.) In this case,

$$
\begin{aligned}
\mathbb{E}\Big[ G_{t_1}[ik]G_{t_2}[jk]G_{\tau_1}[ik]G_{\tau_2}[jk] \Big] &= \mathbb{E}\Big[ G_{t_1}[ik] \Big]\mathbb{E}\Big[ G_{\tau_1}[ik] \Big]\mathbb{E}\Big[ G_{t_2}[jk]G_{\tau_2}[jk] \Big] \\
&= w_{ik}w_{ik}\mathbb{E}\Big[ G_{t_2}[jk]^2 \Big] \\
&= w_{ik}^2 w_{jk}.
\end{aligned}
$$

Substituting this result into (8) yields the covariance

$$
\text{Cov}\Big[ G_{t_1}[ik]G_{t_2}[jk],\ G_{\tau_1}[ik]G_{\tau_2}[jk] \Big] = w_{ik}^2 w_{jk} - w_{ik}^2 w_{jk}^2 = w_{ik}^2 w_{jk}(1 - w_{jk}) \leq 1.
$$

**Case 3**. Assume $\tau_1 = t_1$ but $\tau_2 \neq t_2$. (Occurs $(T-1)T^2$ times.) In this case,

$$
\mathbb{E}\Big[ G_{t_1}[ik]G_{t_2}[jk]G_{\tau_1}[ik]G_{\tau_2}[jk] \Big] = w_{ik}w_{jk}^2,
$$

and so the covariance becomes

$$
\text{Cov}\Big[ G_{t_1}[ik]G_{t_2}[jk],\ G_{\tau_1}[ik]G_{\tau_2}[jk] \Big] = w_{ik}w_{jk}^2(1 - w_{ik}) \leq 1.
$$

Combining all 3 cases, we have the following bound:

$$
\begin{aligned}
\text{Var}[\hat{r}_{ij}^k] &= \frac{1}{T^4}\text{Var}\left[ \sum_{t_1}\sum_{t_2} G_{t_1}[ik]G_{t_2}[jk] \right] \\
&= \frac{1}{T^4}\left[ \sum_{t_1}\sum_{t_2}\text{Var}\Big[ G_{t_1}[ik]G_{t_2}[jk] \Big] + (T-1)T^2 w_{ik}^2 w_{jk}(1 - w_{jk}) + (T-1)T^2 w_{ik}w_{jk}^2(1 - w_{ik}) \right] \\
&= \frac{1}{T^4}\left[ T^2 w_{ik}w_{jk}(1 - w_{ik}w_{jk}) + (T-1)T^2 w_{ik}^2 w_{jk}(1 - w_{jk}) + (T-1)T^2 w_{ik}w_{jk}^2(1 - w_{ik}) \right] \\
&\leq \frac{1}{T^4}\left[ T^2 + 2(T-1)T^2 \right] \\
&= \frac{2T - 1}{T^2} \leq \frac{2}{T}.
\end{aligned}
$$

The bound for $\mathrm{Var}\left[\widehat{c}_{ij}^k\right]$ can be proved similarly.

Next, we observe that $G_t$ (for any $t$) is a directed graph. So the random variables $G_{t_1}[i,k]$ and $G_{t_1}[k,i]$ are independent. Similarly, $G_{t_2}[j,k]$ and $G_{t_2}[k,j]$ are independent. Therefore, the product variables $G_{t_1}[i,k]G_{t_2}[j,k]$ and $G_{t_1}[k,i]G_{t_2}[k,j]$ must be independent for any fixed $u_i$, $u_j$ and $u_k$, where $i \neq j$ and $k = \{1,\ldots,n\}\backslash\{i,j\}$. Consequently, $\widehat{r}_{ij}^k$ and $\widehat{c}_{ij}^k$ are independent, and hence

$$\mathbb{E}[\widehat{r}_{ij}^k\widehat{c}_{ij}^k] = \mathbb{E}\left[\widehat{r}_{ij}^k\right] \cdot \mathbb{E}\left[\widehat{c}_{ij}^k\right]$$
$$= r_{ij}c_{ij},$$

which implies that $\widehat{r}_{ij}^k$ and $\widehat{c}_{ij}^k$ are uncorrelated: $\mathbb{E}\left[(\widehat{r}_{ij}^k - r_{ij})(\widehat{c}_{ij}^k - c_{ij})\right] = 0$. Consequently,

$$\mathrm{Var}\left[\frac{1}{2}\left(\widehat{r}_{ij}^k + \widehat{c}_{ij}^k\right)\right] = \frac{1}{4}\left(\mathrm{Var}\left[\widehat{r}_{ij}^k\right] + \mathrm{Var}\left[\widehat{c}_{ij}^k\right]\right) \leq \frac{1}{T}.$$

Since $\widehat{r}_{ij} = \frac{1}{S}\sum\limits_{k\in\mathcal{S}}\widehat{r}_{ij}^k$ and $\widehat{c}_{ij} = \frac{1}{S}\sum\limits_{k\in\mathcal{S}}\widehat{c}_{ij}^k$, by Bernstein's inequality we have

$$\mathrm{Pr}\left[\left|\frac{1}{2}\left(\widehat{r}_{ij} + \widehat{c}_{ij}\right) - \frac{1}{2}\left(r_{ij} + c_{ij}\right)\right| > \epsilon\right] = \mathrm{Pr}\left[\left|\frac{1}{S}\sum\limits_{k\in\mathcal{S}}\frac{1}{2}\left(\widehat{r}_{ij}^k + \widehat{c}_{ij}^k\right) - \frac{1}{2}\left(r_{ij} + c_{ij}\right)\right| > \epsilon\right]$$

$$\leq 2e^{-\frac{S\epsilon^2}{2\left(\mathrm{Var}\left[\frac{1}{2}\left(\widehat{r}_{ij}^k+\widehat{c}_{ij}^k\right)\right]+\epsilon/3\right)}} \leq 2e^{-\frac{S\epsilon^2}{2(1/T+\epsilon/3)}}.$$

Finally, we note that

$$|\widehat{d}_{ij} - d_{ij}| \leq \frac{1}{2}|\widehat{r}_{ii} + \widehat{c}_{ii} - r_{ii} - c_{ii}| + \frac{1}{2}|\widehat{r}_{ij} + \widehat{c}_{ij} - r_{ij} - c_{ij}| +$$
$$\frac{1}{2}|\widehat{r}_{ji} + \widehat{c}_{ji} - r_{ji} - c_{ji}| + \frac{1}{2}|\widehat{r}_{jj} + \widehat{c}_{jj} - r_{jj} - c_{jj}|.$$

Therefore by union bound we have

$$\mathrm{Pr}[|\widehat{d}_{ij} - d_{ij}| > \epsilon]$$
$$\leq \mathrm{Pr}\left[\frac{1}{2}|\widehat{r}_{ii} + \widehat{c}_{ii} - r_{ii} - c_{ii}| + \frac{1}{2}|\widehat{r}_{ij} + \widehat{c}_{ij} - r_{ij} - c_{ij}| +\right.$$
$$\left.+ \frac{1}{2}|\widehat{r}_{ji} + \widehat{c}_{ji} - r_{ji} - c_{ji}| + \frac{1}{2}|\widehat{r}_{jj} + \widehat{c}_{jj} - r_{jj} - c_{jj}| > \epsilon\right]$$
$$\leq \mathrm{Pr}\left[\left|\frac{1}{2}\left(\widehat{r}_{ii} + \widehat{c}_{ii}\right) - \frac{1}{2}\left(r_{ii} + c_{ii}\right)\right| > \epsilon/4\right] + \mathrm{Pr}\left[\left|\frac{1}{2}\left(\widehat{r}_{ij} + \widehat{c}_{ij}\right) - \frac{1}{2}\left(r_{ij} + c_{ij}\right)\right| > \epsilon/4\right] +$$
$$+ \mathrm{Pr}\left[\left|\frac{1}{2}\left(\widehat{r}_{ji} + \widehat{c}_{ji}\right) - \frac{1}{2}\left(r_{ji} + c_{ji}\right)\right| > \epsilon/4\right] + \mathrm{Pr}\left[\left|\frac{1}{2}\left(\widehat{r}_{jj} + \widehat{c}_{jj}\right) - \frac{1}{2}\left(r_{jj} + c_{jj}\right)\right| > \epsilon/4\right]$$
$$\leq 8e^{-\frac{S\epsilon^2/16}{2(1/T+\epsilon/12)}} = 8e^{-\frac{S\epsilon^2}{32/T+8\epsilon/3}}.$$

If the graph is un-directed, then $c_{ij}^k = r_{ij}^k$ and we can only have $\mathrm{Var}\left[\frac{1}{2}\left(r_{ij}^k + c_{ij}^k\right)\right] \leq \frac{2}{T}$ instead of $\mathrm{Var}\left[\frac{1}{2}\left(r_{ij}^k + c_{ij}^k\right)\right] \leq \frac{1}{T}$. In this case,

$$\mathrm{Pr}[|\widehat{d}_{ij} - d_{ij}| > \epsilon] \leq 8e^{-\frac{S\epsilon^2}{64/T+8\epsilon/3}}.$$

$\square$

# 2 Proofs for Section 3.2

**Theorem 2.** *Let $\Delta$ be the accuracy parameter and $K$ be the number of blocks estimated by Algorithm 1, then*

$$\Pr\left[K > \frac{QL\sqrt{2}}{\Delta}\right] \leq 8n^2 e^{-\frac{S\Delta^4}{128/T+16\Delta^2/3}}, \tag{9}$$

*where $L$ is the Lipschitz constant and $Q$ is the number of Lipschitz blocks in the ground truth $w$.*

*Proof.* Recall that in defining the Lipschitz condition of $w$ (Section 2.1), we defined a sequence of non-overlapping intervals $I_k = [\alpha_k, \alpha_{k+1}]$, where $0 = \alpha_0 < \ldots < \alpha_Q = 1$, and $Q$ is the number of Lipschitz blocks of $w$. For each of the interval $I_k$, we divide it into $R \stackrel{\text{def}}{=} \frac{L\sqrt{2}}{\Delta}$ subintervals of equal size $1/R$. Thus, the distance between any two elements in the same subinterval is at most $1/R$. Also, the total number of subintervals over $[0,1]$ is $QR$.

Now, suppose that there are $K > QR = \frac{QL\sqrt{2}}{\Delta}$ blocks defined by the algorithm, and denote the $K$ pivots be $p_1, \ldots, p_K$. By the pigeonhole principle, there must be at least two pivots $p_i$ and $p_j$ in the same sub-interval. In this case, the distance $d_{p_i, p_j}$ must satisfy the following condition:

$$
\begin{aligned}
d_{p_i,p_j} &= \frac{1}{2}\left(\int_0^1 (w(x,u_{p_i}) - w(x,u_{p_j}))^2 dx + \int_0^1 (w(u_{p_i},y) - w(u_{p_j},y))^2 dy\right)\\
&\leq L^2(u_{p_i} - u_{p_j})^2\\
&\leq L^2 \frac{1}{R^2} = \frac{\Delta^2}{2}.
\end{aligned}
$$

However, from the algorithm it holds that $\widehat{d}_{p_i,p_j} \geq \Delta^2$. So, if $K > QR$, then $\widehat{d}_{p_i,p_j} - d_{p_i,p_j} > \frac{\Delta^2}{2}$.

Let $\mathcal{E}$ be the following event:

$$\mathcal{E} = \left\{\widehat{d}_{p_i,p_j} - d_{p_i,p_j} > \frac{\Delta^2}{2} \quad \text{for at least one pair of } p_i, p_j\right\}.$$

Then, since the event $\mathcal{E}$ is a consequence of the event $\{K > QR\}$, we have

$$\Pr\left[K > \frac{QL\sqrt{2}}{\Delta}\right] = \Pr[K > QR] \leq \Pr[\,\mathcal{E}\,].$$

To bound $\Pr[\mathcal{E}]$, we observe that

$$\Pr\left[\widehat{d}_{p_i,p_j} - d_{p_i,p_j} > \frac{\Delta^2}{2} \,\Big|\, p_i, p_j\right] \leq 8e^{-\frac{S(\Delta^2/2)^2}{32/T+8(\Delta^2/2)/3}} = 8e^{-\frac{S\Delta^4}{128/T+16\Delta^2/3}}.$$

Therefore, by union bound,

$$
\begin{aligned}
\Pr\left[\mathcal{E} \,\Big|\, p_1, \ldots, p_K\right] &\leq \sum_{p_i, p_j} \Pr\left[\widehat{d}_{p_i,p_j} - d_{p_i,p_j} > \frac{\Delta^2}{2} \,\Big|\, p_i, p_j\right]\\
&\leq 8n^2 e^{-\frac{S\Delta^4}{128/T+16\Delta^2/3}},
\end{aligned}
$$

and hence,

$$\Pr[\,\mathcal{E}\,] = \sum_{p_1,\ldots,p_K} \Pr[\mathcal{E}\,|\,p_1,\ldots,p_K]\Pr[p_1,\ldots,p_K]$$

$$\leq \left(8n^2 e^{-\frac{S\Delta^4}{128/T+16\Delta^2/3}}\right) \cdot \sum_{p_1,\ldots,p_K} \Pr[p_1,\ldots,p_K]$$

$$= 8n^2 e^{-\frac{S\Delta^4}{128/T+16\Delta^2/3}}.$$

This completes the proof. $\qquad\qquad\square$

## 3  Proofs for Section 3.3

**Lemma 1.** *Let $\widehat{B}_i = \{i_1, i_2, \ldots, i_{|\widehat{B}_i|}\}$ and $\widehat{B}_j = \{j_1, j_2, \ldots, j_{|\widehat{B}_j|}\}$ be two clusters returned by the Algorithm. Suppose that $\{u_{i_1}, u_{i_2}, \ldots, u_{i_{|\widehat{B}_i|}}\}$ and $\{u_{j_1}, u_{j_2}, \ldots, u_{j_{|\widehat{B}_j|}}\}$ are the ground truth labels of the vertices in $\widehat{B}_i$ and $\widehat{B}_j$, respectively. Let*

$$\overline{w}_{ij} = \frac{1}{|\widehat{B}_i||\widehat{B}_j|} \sum_{i_x \in \widehat{B}_i} \sum_{j_x \in \widehat{B}_j} w(u_{i_x}, u_{j_x}). \tag{10}$$

*Assume that the precision parameter satisfies $\Delta^2 < \frac{\delta^2 L}{4}$, where $L$ is the Lipschitz constant and $\delta$ is the size of the smallest Lipschitz interval. Then, for any $i_v \in \widehat{B}_i$ and $j_v \in \widehat{B}_j$,*

$$\Pr\left[|\overline{w}_{ij} - w(u_{i_v,j_v})| > 8\Delta^{1/2}L^{1/4}\right] \leq 32|\widehat{B}_i||\widehat{B}_j|e^{-\frac{S\Delta^4}{32/T+8\Delta^2/3}}. \tag{11}$$

*Proof.* Let $i_p \in \widehat{B}_i$ and $j_p \in \widehat{B}_j$ be pivots of the clusters $\widehat{B}_i$ and $\widehat{B}_j$, respectively. By definition of pivots, it holds that $|\widehat{d}_{i_p,i_v}| \leq \Delta^2$ and $|\widehat{d}_{j_p,j_v}| \leq \Delta^2$ for any vertices $i_v \in \widehat{B}_i$ and $j_v \in \widehat{B}_j$. Therefore,

$$\begin{aligned} 0 &\leq -|\widehat{d}_{i_p,i_v}| + \Delta^2 \leq -\widehat{d}_{i_p,i_v} + \Delta^2 \\ \Rightarrow \quad d_{i_p,i_v} &\leq d_{i_p,i_v} - \widehat{d}_{i_p,i_v} + \Delta^2 \leq |d_{i_p,i_v} - \widehat{d}_{i_p,i_v}| + \Delta^2, \end{aligned}$$

which implies that

$$\begin{aligned} \Pr\left[d_{i_p,i_v} > 2\Delta^2\right] &\leq \Pr\left[|d_{i_p,i_v} - \widehat{d}_{i_p,i_v}| + \Delta^2 > 2\Delta^2\right] \\ &= \Pr\left[|d_{i_p,i_v} - \widehat{d}_{i_p,i_v}| > \Delta^2\right] \\ &\leq 8e^{-\frac{S\Delta^4}{32/T+8\Delta^2/3}}. \end{aligned}$$

Similarly, we have $\Pr\left[d_{j_p,j_v} > 2\Delta^2\right] \leq 8e^{-\frac{S\Delta^4}{32/T+8\Delta^2/3}}$. Thus,

$$\begin{aligned} \Pr\left[d_{i_p,i_v} > 2\Delta^2 \,\cup\, d_{j_p,j_v} > 2\Delta^2\right] &\leq \Pr\left[d_{i_p,i_v} > 2\Delta^2\right] + \Pr\left[d_{j_p,j_v} > 2\Delta^2\right] \\ &\leq 16e^{-\frac{S\Delta^4}{32/T+8\Delta^2/3}}. \end{aligned}$$

Let $d_{ij}^c = \int_0^1 (w(x, u_i) - w(x, u_j))^2 dx$ and $d_{ij}^r = \int_0^1 (w(u_i, y) - w(u_j, y))^2 dy$. By Lemma 5, it holds that for any $0 < (\epsilon/2)^2 < 2\delta L$, if $d_{i,j}^c \leq \frac{(\epsilon/2)^4}{8L} = \frac{\epsilon^4}{128L}$ and $d_{i,j}^r \leq \frac{\epsilon^4}{128L}$, then

$$\sup_{x \in [0,1]} |w(x, u_i) - w(x, u_j)| \leq \frac{\epsilon}{2},$$

$$\sup_{y \in [0,1]} |w(u_i, y) - w(u_j, y)| \leq \frac{\epsilon}{2}.$$

Therefore, if $d_{i_p,i_v}^c \leq \frac{\epsilon^4}{128L}$, $d_{i_p,i_v}^r \leq \frac{\epsilon^4}{128L}$, $d_{j_p,j_v}^c \leq \frac{\epsilon^4}{128L}$ and $d_{j_p,j_v}^r \leq \frac{\epsilon^4}{128L}$, then for pivots $i_p \in \widehat{B}_i$, $j_p \in \widehat{B}_j$, and vertex $i_v \in \widehat{B}_i$, $j_v \in \widehat{B}_j$:

$$
\begin{aligned}
|w(u_{i_v}, u_{j_v}) - w(u_{i_p}, u_{j_p})| &\leq |w(u_{i_v}, u_{j_v}) - w(u_{i_v}, u_{j_p})| + |w(u_{i_v}, u_{j_p}) - w(u_{i_p}, u_{j_p})| \\
&\leq \sup_{x \in [0,1]} |w(x, u_{j_v}) - w(x, u_{j_p})| + \sup_{y \in [0,1]} |w(u_{i_v}, y) - w(u_{j_p}, y)| \\
&= \frac{\epsilon}{2} + \frac{\epsilon}{2} = \epsilon.
\end{aligned}
\tag{12}
$$

Also, if $d_{i_p,i_x}^c \leq \frac{\epsilon^4}{128L}$, $d_{i_p,i_x}^r \leq \frac{\epsilon^4}{128L}$, $d_{j_p,j_x}^c \leq \frac{\epsilon^4}{128L}$ and $d_{j_p,j_x}^r \leq \frac{\epsilon^4}{128L}$ for vertex every $i_x \in \widehat{B}_i$, $j_x \in \widehat{B}_j$

$$
\begin{aligned}
&\left| \frac{1}{|\widehat{B}_i||\widehat{B}_j|} \sum_{i_x \in \widehat{B}_i} \sum_{j_x \in \widehat{B}_j} w(u_{i_x}, u_{j_x}) - w(u_{i_p}, u_{j_p}) \right| \\
&\leq \left| \frac{1}{|\widehat{B}_i||\widehat{B}_j|} \sum_{i_x \in \widehat{B}_i} \sum_{j_x \in \widehat{B}_j} w(u_{i_x}, u_{j_x}) - \frac{1}{|\widehat{B}_i|} \sum_{i_x \in \widehat{B}_i} w(u_{i_x}, u_{j_p}) \right| + \left| \frac{1}{|\widehat{B}_i|} \sum_{i_x \in \widehat{B}_i} w(u_{i_x}, u_{j_p}) - w(u_{i_p}, u_{j_p}) \right| \\
&\leq \frac{1}{|\widehat{B}_i|} \frac{1}{|\widehat{B}_j|} \sum_{i_x \in \widehat{B}_i} \sum_{j_x \in \widehat{B}_j} |w(u_{i_x}, u_{j_x}) - w(u_{i_x}, u_{j_p})| + \frac{1}{|\widehat{B}_i|} \sum_{i_x \in \widehat{B}_i} |w(u_{i_x}, u_{j_p}) - w(u_{i_p}, u_{j_p})| \\
&\leq \frac{1}{|\widehat{B}_i|} \frac{1}{|\widehat{B}_j|} \sum_{i_x \in \widehat{B}_i} \sum_{j_x \in \widehat{B}_j} \frac{\epsilon}{2} + \frac{1}{|\widehat{B}_i|} \sum_{i_x \in \widehat{B}_i} \frac{\epsilon}{2} = \epsilon.
\end{aligned}
\tag{13}
$$

Combining (12) and (13) with triangle inequality yields

$$\left| \frac{1}{|\widehat{B}_i||\widehat{B}_j|} \sum_{i_x \in \widehat{B}_i} \sum_{j_x \in \widehat{B}_j} w(u_{i_x}, u_{j_x}) - w(u_{i_v}, u_{j_v}) \right| \leq 2\epsilon.$$

Consequently, by contrapositive this implies that

$$
\begin{aligned}
&|\overline{w}_{ij} - w(u_{i_v}, u_{j_v})| > 2\epsilon \\
&\Rightarrow \bigcup_{i_x \in \widehat{B}_i, j_x \in \widehat{B}_j} \left( d_{i_p,i_x}^c > \frac{\epsilon^4}{128L} \cup d_{i_p,i_x}^r > \frac{\epsilon^4}{128L} \cup d_{j_p,j_x}^c > \frac{\epsilon^4}{128L} \cup d_{j_p,j_x}^r > \frac{\epsilon^4}{128L} \right) \\
&\Rightarrow \bigcup_{i_x \in \widehat{B}_i, j_x \in \widehat{B}_j} \left( d_{i_p,i_x} > \frac{\epsilon^4}{128L} \cup d_{j_p,j_x} > \frac{\epsilon^4}{128L} \right).
\end{aligned}
$$

Therefore,

$$\Pr\left[|\overline{w}_{ij} - w(u_{i_v}, u_{j_v})| > 2\epsilon\right] \leq \Pr\left[\bigcup_{i_x \in \widehat{B}_i, j_x \in \widehat{B}_j} \left(d_{i_p, i_x} > \frac{\epsilon^4}{128L} \cup d_{j_p, j_x} > \frac{\epsilon^4}{128L}\right)\right]$$

$$\leq \sum_{i_x \in \widehat{B}_i, j_x \in \widehat{B}_j} \left(\Pr\left[d_{i_p, i_x} > \frac{\epsilon^4}{128L}\right] + \Pr\left[d_{j_p, j_x} > \frac{\epsilon^4}{128L}\right]\right).$$

Assuming $\Delta < \delta\sqrt{L}/2$ and setting $\epsilon = 4\Delta^{1/2}L^{1/4}$, we have $0 < (\epsilon/2)^2 < 2\delta L$ and thus

$$\Pr\left[|\overline{w}_{ij} - w(u_{i_v}, u_{j_v})| > 8\Delta^{1/2}L^{1/4}\right] \leq \sum_{i_x \in \widehat{B}_i, j_x \in \widehat{B}_j} \left(\Pr\left[d_{i_p, i_x} > 2\Delta^2\right] + \Pr\left[d_{j_p, j_x} > 2\Delta^2\right]\right)$$

$$\leq 32|\widehat{B}_i||\widehat{B}_j|e^{-\frac{S\Delta^4}{32/T + 8\Delta^2/3}}.$$

$\square$

**Lemma 2.** *Let $\widehat{B}_i = \{i_1, i_2, \ldots, i_{|\widehat{B}_i|}\}$ and $\widehat{B}_j = \{j_1, j_2, \ldots, j_{|\widehat{B}_j|}\}$ be two clusters returned by the Algorithm. Suppose that $\{u_{i_1}, u_{i_2}, \ldots, u_{i_{|\widehat{B}_i|}}\}$ and $\{u_{j_1}, u_{j_2}, \ldots, u_{j_{|\widehat{B}_j|}}\}$ are the ground truth labels of the vertices in $\widehat{B}_i$ and $\widehat{B}_j$, respectively. Let*

$$\widehat{w}_{ij} = \frac{1}{|\widehat{B}_i||\widehat{B}_j|} \sum_{i_x \in \widehat{B}_i} \sum_{j_x \in \widehat{B}_j} \left(\frac{G_1[i_x, j_x] + \ldots + G_{2T}[i_x, j_x]}{2T}\right),$$

$$\overline{w}_{ij} = \frac{1}{|\widehat{B}_i||\widehat{B}_j|} \sum_{i_x \in \widehat{B}_i} \sum_{j_x \in \widehat{B}_j} w(u_{i_x}, u_{j_x}).$$

*Then,*

$$\Pr\left[|\widehat{w}_{ij} - \overline{w}_{ij}| > 8\Delta^{1/2}L^{1/4}\right] \leq 2e^{-256(T|\widehat{B}_i||\widehat{B}_j|\sqrt{L}\Delta)} + 32|\widehat{B}_i|^2|\widehat{B}_j|^2 e^{-\frac{S\Delta^4}{32/T + 8\Delta^2/3}}.$$

*Proof.* There are two possible situations that we need to consider.

**Case 1**: For any vertex $i_v \in \widehat{B}_i$ and $j_v \in \widehat{B}_j$, the estimate of the previous lemma $\overline{w}_{ij}$ (independent of $(i_v, j_v)$) is close to the ground truth $w_{ij} \overset{\text{def}}{=} w(u_{i_v}, u_{j_v})$. In other words, we want $w(u_{i_v}, u_{j_v})$ to stay close for all $i_v \in \widehat{B}_i$ and $j_v \in \widehat{B}_j$, so that the difference $|w_{ij} - \overline{w}_{ij}|$ remains small for all $i_v \in \widehat{B}_i$ and $j_v \in \widehat{B}_j$.

**Case 2**: Complement of case 1.

To encapsulate these two cases, we first define the event

$$\mathcal{E} = \left\{|w_{ij} - \overline{w}_{ij}| \leq 8\Delta^{1/2}L^{1/4}, \forall i_v \in \widehat{B}_i, \ j_v \in \widehat{B}_j\right\}$$

and define $\overline{\mathcal{E}}$ be the complement of $\mathcal{E}$. Then,

$$
\begin{aligned}
\Pr\left[|\widehat{w}_{ij} - \overline{w}_{ij}| > 8\Delta^{1/2}L^{1/4}\right] = {} & \Pr\left[|\widehat{w}_{ij} - \overline{w}_{ij}| > 8\Delta^{1/2}L^{1/4} \,\middle|\, \mathcal{E}\right]\Pr\left[\mathcal{E}\right] \\
& + \Pr\left[|\widehat{w}_{ij} - \overline{w}_{ij}| > 8\Delta^{1/2}L^{1/4} \,\middle|\, \overline{\mathcal{E}}\right]\Pr\left[\overline{\mathcal{E}}\right] \\
\leq {} & \Pr\left[|\widehat{w}_{ij} - \overline{w}_{ij}| > 8\Delta^{1/2}L^{1/4} \,\middle|\, \mathcal{E}\right] + \Pr\left[\overline{\mathcal{E}}\right].
\end{aligned}
$$

So it remains to bound the two probabilities.

Conditioning on $\mathcal{E}$, it holds that

$$
\overline{w}_{ij} - \epsilon \leq w_{ij} \leq \overline{w} + \epsilon.
$$

Fix a vertex pair $(i_v, j_v)$, we note that $G_1[i_v, j_v], \ldots, G_{2T}[i_v, j_v]$ are independent Bernoulli random variable with common mean $w(u_{i_v}, u_{j_v})$. Denote

$$
\widehat{w}_{ij} = \frac{1}{2T|\widehat{B}_i||\widehat{B}_j|}\sum_{t=1}^{2T}\sum_{i_x \in \widehat{B}_i}\sum_{j_x \in \widehat{B}_j}G_t[i_x, j_x],
$$

then by Hoeffding inequality we have

$$
\begin{aligned}
\Pr\left[\widehat{w}_{ij} - \overline{w}_{ij} > 2\epsilon \,\middle|\, \mathcal{E}\right] = {} & \Pr\left[\widehat{w}_{ij} > \overline{w}_{ij} + 2\epsilon \,\middle|\, \mathcal{E}\right] \\
\leq {} & \Pr\left[\widehat{w}_{ij} > w_{ij} + \epsilon \,\middle|\, \mathcal{E}\right] \\
\leq {} & e^{-2(2T|\widehat{B}_i||\widehat{B}_j|\epsilon^2)},
\end{aligned}
$$

and similarly $\Pr\left[\widehat{w}_{ij} - \overline{w}_{ij} < -2\epsilon \,\middle|\, \mathcal{E}\right] \leq e^{-2(2T|\widehat{B}_i||\widehat{B}_j|\epsilon^2)}$. Therefore,

$$
\Pr\left[|\widehat{w}_{ij} - \overline{w}_{ij}| > 2\epsilon \,\middle|\, \mathcal{E}\right] \leq 2e^{-2(2T|\widehat{B}_i||\widehat{B}_j|\epsilon^2)}.
$$

Substituting $\epsilon = 4\Delta^{1/2}L^{1/4}$, we have

$$
\Pr\left[|\widehat{w}_{ij} - \overline{w}_{ij}| > 8\Delta^{1/2}L^{1/4} \,\middle|\, \mathcal{E}\right] \leq 2e^{-128(|\widehat{B}_i||\widehat{B}_j|(2T)\sqrt{L}\Delta)}.
$$

The second probability is bounded as follows. Since $\overline{\mathcal{E}}$ is the complement of $\mathcal{E}$, it is bounded by the probability where at least one $(i_v, j_v)$ violates the condition. Therefore,

$$
\begin{aligned}
\Pr\left[\overline{\mathcal{E}}\right] = {} & \Pr\left[\text{at least one } i_v, j_v \text{ s.t. } |w(u_{i_v}, u_{j_v}) - \overline{w}_{ij}| > 8\Delta^{1/2}L^{1/4}\right] \\
\leq {} & \sum_{i_v \in \widehat{B}_i}\sum_{j_v \in \widehat{B}_j}\Pr\left[|w(u_{i_v}, u_{j_v}) - \overline{w}_{ij}| > 8\Delta^{1/2}L^{1/4}\right] \\
\leq {} & 32|\widehat{B}_i|^2|\widehat{B}_j|^2 e^{-\frac{S\Delta^4}{32/T + 8\Delta^2/3}}.
\end{aligned}
$$

Finally, by combining the above results we have

$$
\Pr\left[|\widehat{w}_{ij} - \overline{w}_{ij}| > 8\Delta^{1/2}L^{1/4}\right] \leq 2e^{-256(T|\widehat{B}_i||\widehat{B}_j|\sqrt{L}\Delta)} + 32|\widehat{B}_i|^2|\widehat{B}_j|^2 e^{-\frac{S\Delta^4}{32/T + 8\Delta^2/3}}.
$$

$\square$

**Lemma 3.** *Let $\widehat{B}_i = \{i_1, i_2, \ldots, i_{|\widehat{B}_i|}\}$ and $\widehat{B}_j = \{j_1, j_2, \ldots, j_{|\widehat{B}_j|}\}$ be two clusters returned by the Algorithm. Suppose that $\{u_{i_1}, u_{i_2}, \ldots, u_{i_{|\widehat{B}_i|}}\}$ and $\{u_{j_1}, u_{j_2}, \ldots, u_{j_{|\widehat{B}_j|}}\}$ are the ground truth labels of the vertices in $\widehat{B}_i$ and $\widehat{B}_j$, respectively. Let*

$$\widehat{w}_{ij} = \frac{1}{|\widehat{B}_i||\widehat{B}_j|} \sum_{i_x \in \widehat{B}_i} \sum_{j_x \in \widehat{B}_j} \left( \frac{G_1[i_x, j_x] + \ldots + G_{2T}[i_x, j_x]}{2T} \right).$$

*Then,*

$$\Pr\left[ |\widehat{w}_{ij} - w_{ij}| > 16\Delta^{1/2}L^{1/4} \right] \leq 2e^{-256(T|\widehat{B}_i||\widehat{B}_j|\sqrt{L}\Delta)} + 64n^4 e^{-\frac{S\Delta^4}{32/T+8\Delta^2/3}}.$$

*Proof.* By Lemma 1 and Lemma 2, we have

$$\Pr\left[ |\widehat{w}_{ij} - \overline{w}_{ij}| > 8\Delta^{1/2}L^{1/4} \right] \leq 2e^{-256(T|\widehat{B}_i||\widehat{B}_j|\sqrt{L}\Delta)} + 32|\widehat{B}_i|^2|\widehat{B}_j|^2 e^{-\frac{S\Delta^4}{32/T+8\Delta^2/3}}$$

$$\Pr\left[ \overline{w}_{ij} - w_{ij} > 8\Delta^{1/2}L^{1/4} \right] \leq 32|\widehat{B}_i||\widehat{B}_j| e^{-\frac{S\Delta^4}{32/T+8\Delta^2/3}}.$$

Therefore, it follows that

$$\Pr\left[ |\widehat{w}_{ij} - w_{ij}| > 16\Delta^{1/2}L^{1/4} \right] \leq \Pr\left[ |\widehat{w}_{ij} - \overline{w}_{ij}| > 8\Delta^{1/2}L^{1/4} \right] + \Pr\left[ \overline{w}_{ij} - w_{ij} > 8\Delta^{1/2}L^{1/4} \right]$$

$$\leq 2e^{-256(T|\widehat{B}_i||\widehat{B}_j|\sqrt{L}\Delta)} + 32|\widehat{B}_i|^2|\widehat{B}_j|^2 e^{-\frac{S\Delta^4}{32/T+8\Delta^2/3}} + 32|\widehat{B}_i||\widehat{B}_j| e^{-\frac{S\Delta^4}{32/T+8\Delta^2/3}}$$

$$\leq 2e^{-256(T|\widehat{B}_i||\widehat{B}_j|\sqrt{L}\Delta)} + 64n^4 e^{-\frac{S\Delta^4}{32/T+8\Delta^2/3}}.$$

$\square$

**Lemma 4.** *Let $E$ be a subset of the edge set $E_0 = \{(i,j) \mid i \in \{1, \ldots, n\}, j \in \{1, \ldots, n\}\}$. Then under the above setup, there exists constants $c_0$ and $c_1$ such that*

$$\Pr\left[ \frac{1}{|E|} \sum_{i_v, j_v \in E} |w(u_{i_v}, u_{j_v}) - \widehat{w}_{ij}| > c_0\sqrt{\Delta} \right] \leq \sum_{i_v, j_v \in E} 2e^{-c_1(T|\widehat{B}_i||\widehat{B}_j|\Delta)} + 64|E|n^4 e^{-\frac{S\Delta^4}{32/T+8\Delta^2/3}}. \quad (14)$$

*Proof.* From Lemma 3, average over all pairs $(i_v, j_v) \in E$,

$$\Pr\left[ \frac{1}{|E|} \sum_{i_v, j_v \in E} |w(u_{i_v}, u_{j_v}) - \widehat{w}_{ij}| > 16\Delta^{1/2}L^{1/4} \right] \leq \frac{1}{|E|} \sum_{i_v, j_v \in E} \Pr\left[ |w(u_{i_v}, u_{j_v}) - \widehat{w}_{ij} > 16\Delta^{1/2}L^{1/4}| \right]$$

$$\leq \sum_{i_v, j_v \in E} 2e^{-256(T|\widehat{B}_i||\widehat{B}_j|\sqrt{L}\Delta)} + 64|E|n^4 e^{-\frac{S\Delta^4}{32/T+8\Delta^2/3}}.$$

Choosing $c_0 = 16L^{1/4}$ and $c_1 = 256\sqrt{L}$ yields the desired result. $\square$

**Lemma 5.** *Let $I_k = [\alpha_{k-1}, \alpha_k]$ for $k = 1, \ldots, K$ be a sequence of intervals such that $I_i \cap I_j = \emptyset$ and $\cup I_i = [0, 1]$. Suppose that $w$ is piecewise Lipschitz continuous and differentiable in $I_k$. For any $u_i, u_j \in [0, 1]$, define*

$$f_{ij}(x) = (w(x, u_i) - w(x, u_j))^2$$
$$g_{ij}(y) = (w(u_i, y) - w(u_j, y))^2,$$

*and*

$$h_{ij}(x, y) = \frac{1}{2} \left[ f_{ij}(x) + g_{ij}(y) \right].$$

*Let $\delta = \min\limits_{k=1,\ldots,K} |\alpha_k - \alpha_{k-1}|$. If*

$$d_{ij}^c = \int_0^1 f_{ij}(x) dx \leq \frac{\epsilon^2}{8L}, \qquad and \qquad d_{ij}^r = \int_0^1 g_{ij}(y) dy \leq \frac{\epsilon^2}{8L},$$

*for some constant $0 < \epsilon < 2\delta L$, then*

$$\sup_{x \in [0,1]} f_{ij}(x) \leq \epsilon, \qquad and \qquad \sup_{y \in [0,1]} g_{ij}(y) \leq \epsilon.$$

*Hence, $\sup\limits_{(x,y) \in [0,1]^2} h_{ij}(x, y) \leq \epsilon.$*

*Proof.* Since $h_{ij}(x, y)$ is separable, it is sufficient to prove for $f_{ij}(x)$.

Fix $i$ and $j$, and let $f_{ij}^{sup} = \sup\limits_{x \in [0,1]} f_{ij}(x)$. Let $I_k = [\alpha_{k-1}, \alpha_k]$ be the interval such that $f_{ij}^{sup}$ is attained, and let $\delta_k = |\alpha_k - \alpha_{k-1}|$ be the width of the interval. Consider a neighborhood surrounding the center of $I_k$ with radius $\delta_k/2 - \theta$, where $0 < \theta < \delta_k/2$. Then define

$$f_{ij}^{sup}(\theta) = \sup_{x \in [\alpha_{k-1}+\theta, \alpha_k-\theta]} f_{ij}(x).$$

It is clear that $f_{ij}^{sup} = \lim\limits_{\theta \to 0} f_{ij}^{sup}(\theta)$.

The set $[\alpha_{k-1} + \theta, \alpha_k - \theta]$ is compact, so there exists $x_{ij}^{max}(\theta) \in [\alpha_{k-1} + \theta, \alpha_k - \theta]$ such that $f_{ij}^{sup} = f_{ij}(x_{ij}^{max})$. Assume, without lost of generality, that $x_{ij}^{max}(\theta) + \delta_k/2 - \theta$ (i.e., $x_{ij}^{max}$ is in the lower half of the interval). For any $0 < \epsilon_0 < \frac{\epsilon}{4L} - \theta \leq \frac{\delta}{2} - \theta \leq \frac{\delta_k}{2} - \theta$,

$$\frac{h_{ij}(x_{ij}^{max}(\theta)) - h_{ij}(x_{ij}^{max}(\theta) + \epsilon_0)}{\epsilon_0} =$$

$$\frac{(w(i, x_{ij}^{max}) - w(j, x_{ij}^{max}))^2 - (w(i, x_{ij}^{max}(\theta) + \epsilon_0) - w(j, x_{ij}^{max}(\theta) + \epsilon_0))^2}{\epsilon_0} \leq$$

$$\frac{(w(i, x_{ij}^{max}) - w(j, x_{ij}^{max}))^2 - (w(i, x_{ij}^{max}) + L\epsilon_0 - w(j, x_{ij}^{max}) + L\epsilon_0)^2}{\epsilon_0} \leq$$

$$4L(w(j, x_{ij}^{max}) - w(i, x_{ij}^{max})) \leq 4L \Rightarrow$$

$$\frac{f_{ij}(x_{ij}^{max}(\theta)) - f_{ij}(x_{ij}^{max}(\theta) + \epsilon_0)}{\epsilon_0} \leq 4L,$$

which implies that

$$f_{ij}(x_{ij}^{max}(\theta)) - 4L\epsilon_0 \leq f_{ij}(x_{ij}^{max}(\theta) + \epsilon_0).$$

Integrating both sides with respect to $\epsilon_0$ with limits 0 and $\frac{\epsilon}{4L} - \theta$ yields

$$f_{ij}(x_{ij}^{max}(\theta))\left(\frac{\epsilon}{4L} - \theta\right) - \frac{4L}{2}\left(\frac{\epsilon}{4L} - \theta\right)^2 \leq \int_0^{\frac{\epsilon}{4L} - \theta} f_{ij}(x_{ij}^{max}(\theta) + \epsilon_0)d\epsilon_0$$

$$\leq \int_0^1 f_{ij}(x)dx = d_{ij}^c.$$

Therefore,

$$f_{ij}(x_{ij}^{max}(\theta)) \leq \frac{d_{ij}^c}{\frac{\epsilon}{4L} - \theta} + 2L\left(\frac{\epsilon}{4L} - \theta\right),$$

and hence

$$f_{ij}^{sup} = \lim_{\theta \to 0} f_{ij}^{sup}(\theta) = \lim_{\theta \to 0} f_{ij}(x_{ij}^{max}(\theta)) \leq \frac{4Ld_{ij}^c}{\epsilon} + \frac{\epsilon}{2}.$$

It then follows that if $d_{ij}^c \leq \frac{\epsilon^2}{8L}$, then $f_{ij}^{sup} \leq \epsilon$. $\qquad \square$

**Definition 1.** *The mean squared error (MSE) and mean absolute error (MAE) are defined as*

$$\text{MSE}(\widehat{w}) = \frac{1}{n^2} \sum_{i_v=1}^n \sum_{j_v=1}^n \left(w(u_{i_v}, u_{j_v}) - \widehat{w}_{i_v, j_v}\right)^2 \tag{15}$$

$$\text{MAE}(\widehat{w}) = \frac{1}{n^2} \sum_{i_v=1}^n \sum_{j_v=1}^n \left|w(u_{i_v}, u_{j_v}) - \widehat{w}_{i_v, j_v}\right|. \tag{16}$$

**Theorem 3.** *If $S \in \Theta(n)$ and $\Delta_n \in \omega\left(\left(\frac{\log(n)}{n}\right)^{\frac{1}{4}}\right) \cap o(1)$, then*

$$\lim_{n \to \infty} \mathbb{E}[\text{MAE}(\widehat{w})] = 0 \qquad and \qquad \lim_{n \to \infty} \mathbb{E}[\text{MSE}(\widehat{w})] = 0. \tag{17}$$

*Proof.* Suppose that the algorithm is executed for a set of observed graphs with $n$ vertices using parameters $\Delta_n$ and $S$. Let $K_n'$ be the number of blocks generated. Assume that, as $n \to \infty$, the parameters satisfy $S \in \Theta(n)$ and $\Delta_n \in \omega\left(\left(\frac{\log(n)}{n}\right)^{\frac{1}{4}}\right) \cap o(1)$.

The proof is based on (4). The intuition is to that that the two terms $\sum_{i_v, j_v \in E} 2e^{-c_1(T|\widehat{B}_i||\widehat{B}_j|\Delta)}$ and $32|E|n^4 e^{-\frac{S\Delta^4}{16/T + 8\Delta^2/3}}$ vanish as $n \to \infty$. The latter is clear if $S \in \Theta(n)$ and $\Delta_n \in \omega\left(\left(\frac{\log(n)}{n}\right)^{\frac{1}{4}}\right) \cap$

$o(1)$. For the first term, it is necessary to consider the size $|E|$, which is the size of the cluster generated. We show that the number of small clusters is asymptotically irrelevant. Most of the error come from vertices whose cluster is large enough to make $e^{-\frac{S\Delta^4}{32/T+8\Delta^2/3}}$ vanish.

From Theorem 2, we have

$$\Pr\left[K' > \frac{QL\sqrt{2}}{\Delta_n}\right] \le 8n^2 e^{-\frac{S\Delta_n^4}{128/T+16\Delta_n^2/3}}.$$

Let $\mathcal{E}_n$ be the event that $K'_n \le QL\sqrt{2}/\Delta_n$. Then $\lim_{n\to\infty}\Pr[\mathcal{E}_n] = 1$.

Suppose $\mathcal{E}_n$ happens and define $r_n$ as the number of blocks with less than $\frac{n\Delta_n^2}{QL\sqrt{2}}$ elements. Let $V_n$ be the union of these blocks, and define $\overline{V}_n$ be the complement of $V_n$. Then,

$$|V_n| \le r_n \frac{n\Delta_n^2}{QL\sqrt{2}} \le K'_n \frac{n\Delta_n^2}{QL\sqrt{2}} \le n\Delta_n.$$

So, $|V_n|/n \le \Delta_n$.

Now, let's consider MAE.

$$
\begin{aligned}
\text{MAE} &= \frac{1}{n^2}\sum_{i_v\in V}\sum_{j_v\in V}|w(u_{i_v},u_{j_v})-\widehat{w}_{i_v,j_v}| \\
&= \frac{1}{n^2}\sum_{i_v\in V_n}\sum_{j_v\in V_n}|w(u_{i_v},u_{j_v})-\widehat{w}_{i_v,j_v}| + \frac{1}{n^2}\sum_{i_v\in\overline{V}_n}\sum_{j_v\in\overline{V}_n}|w(u_{i_v},u_{j_v})-\widehat{w}_{i_v,j_v}| + \\
&\quad + \frac{1}{n^2}\sum_{i_v\in\overline{V}_n}\sum_{j_v\in V_n}|w(u_{i_v},u_{j_v})-\widehat{w}_{i_v,j_v}| + \frac{1}{n^2}\sum_{i_v\in V_n}\sum_{j_v\in\overline{V}_n}|w(u_{i_v},u_{j_v})-\widehat{w}_{i_v,j_v}| \\
&\le \frac{|V_n|^2}{n^2} + \frac{|V_n|}{n}\frac{|\overline{V}_n|}{n} + \frac{|\overline{V}_n|}{n}\frac{|V_n|}{n} + \frac{1}{n^2}\sum_{i_v\in\overline{V}_n}\sum_{j_v\in\overline{V}_n}|w(u_{i_v},u_{j_v})-\widehat{w}_{i_v,j_v}| \\
&\le \frac{1}{n^2}\sum_{i_v\in\overline{V}_n}\sum_{j_v\in\overline{V}_n}|w(u_{i_v},u_{j_v})-\widehat{w}_{i_v,j_v}| + \Delta_n^2 + 2\Delta_n \\
&\le \frac{1}{n^2}\sum_{i_v\in\overline{V}_n}\sum_{j_v\in\overline{V}_n}|w(u_{i_v},u_{j_v})-\widehat{w}_{i_v,j_v}| + 3\Delta_n.
\end{aligned}
$$

Similar result holds for MSE:

$$\text{MSE} = \frac{1}{n^2}\sum_{i_v\in V}\sum_{j_v\in V}(w(u_{i_v},u_{j_v})-\widehat{w}_{i_v,j_v})^2 \le \frac{1}{n^2}\sum_{i_v\in\overline{V}_n}\sum_{j_v\in\overline{V}_n}(w(u_{i_v},u_{j_v})-\widehat{w}_{i_v,j_v})^2 + 3\Delta_n.$$

Therefore, using Lemma 4 with $E = \overline{V}_n$:

$$\Pr\left[\text{MAE}(\widehat{w}) > c_0\sqrt{\Delta_n} + 3\Delta_n \;\middle|\; \mathcal{E}\right] \leq \Pr\left[\frac{1}{n^2}\sum_{i_v \in \overline{V}_n}\sum_{j_v \in \overline{V}_n}|w(u_{i_v}, u_{j_v}) - \widehat{w}_{i_v,j_v}| + 3\Delta_n > c_0\sqrt{\Delta_n} + 3\Delta_n \;\middle|\; \mathcal{E}\right]$$

$$\leq \frac{1}{\Pr[\mathcal{E}]}\Pr\left[\frac{1}{|\overline{V}_n|^2}\sum_{i_v \in \overline{V}_n}\sum_{j_v \in \overline{V}_n}|w(u_{i_v}, u_{j_v}) - \widehat{w}_{i_v,j_v}| > c_0\sqrt{\Delta_n} \;\middle|\; \mathcal{E}\right]$$

$$\leq \frac{1}{\Pr[\mathcal{E}]}\left(\sum_{i_v \in \overline{V}_n}\sum_{j_v \in \overline{V}_n}2e^{-256(T|\widehat{B}_i||\widehat{B}_j|\sqrt{L}\Delta)} + 64|\overline{V}_n|n^4 e^{-\frac{S\Delta^4}{32/T + 8\Delta^2/3}}\right).$$

and

$$\Pr\left[\text{MSE}(\widehat{w}) > c_0\sqrt{\Delta_n} + 3\Delta_n \;\middle|\; \mathcal{E}\right] \leq \frac{1}{\Pr[\mathcal{E}]}\left(\sum_{i_v \in \overline{V}_n}\sum_{j_v \in \overline{V}_n}2e^{-256(T|\widehat{B}_i||\widehat{B}_j|\sqrt{L}\Delta)} + 64|\overline{V}_n|n^4 e^{-\frac{S\Delta^2}{32/T + 8\Delta/3}}\right).$$

So,

$$\lim_{n \to \infty}\Pr\left[\text{MAE}(\widehat{w}) > c_0\sqrt{\Delta_n} + 3\Delta_n \;\middle|\; \mathcal{E}\right]\Pr[\mathcal{E}] = 0.$$

Since $\lim_{n \to \infty}\Delta_n = 0$ and $\lim_{n \to \infty}\Pr[\mathcal{E}_n] = 1$, it holds that for any $\epsilon > 0$,

$$\lim_{n \to \infty}\Pr[\text{MAE}(\widehat{w}) > \epsilon] = 0.$$

Finally, since $\widehat{w}$ is bounded in $[0, 1]$,

$$\mathbb{E}[\text{MAE}(\widehat{w})] \leq \epsilon\Pr[\text{MAE}(\widehat{w}) \leq \epsilon] + \Pr[\text{MAE}(\widehat{w}) > \epsilon].$$

Sending $\epsilon \to \infty$,

$$\lim_{n \to \infty}\mathbb{E}[\text{MAE}(\widehat{w})] \leq \lim_{n \to \infty}\Pr[\text{MAE}(\widehat{w}) > \epsilon] = 0.$$

Same arguments hold for MSE. $\qquad\square$