[Reviews · NeurIPS 2013]

Submitted by Assigned_Reviewer_4

This paper introduces a new estimator for graphons - limit objects of a convergent sequence of graphs/network - which takes the form of a network blockmodel. The authors demonstrate the consistency of this estimator and briefly demonstrate the empirical performance of this estimator.

The paper is mostly clear and well written. The theory seems to be mostly sound (see one question below) and the experiments yield some insight into the estimator.

Frequentist estimation of graphons is only recently entering the literature - in this sense the work is original. The first submitted manuscript contained no reference to Choi and Wolfe, Co-clustering separately exchangeable network data, arXiv:1212.4093. This work also demonstrates consistency of blockmodel estimators of graphons. I would expect to see some comparison or comment on this work upon publication.

In the first submitted manuscript the authors did not mention that graphons are unique up to measure preserving transformations of their input variables. This means that one must estimate the equivalence class or perhaps a particular element of the equivalence class. This difficulty is one of the main reasons that consistency results for graphon estimation are only just entering the literature. I hope to see the authors explicitly describe how their estimator is immune to this difficulty upon publication.

It may be instructive to also motivate the estimation of graphons from exchangeability theory as well as graph limits (e.g. Aldous, Hoover, Kallenberg, the connection between the two by Diaconis and Janson, and applications by Hoff, Roy and Teh, Lloyd et alia)

Minor comments
- 41 - this seems like an unusual way to phrase the difference between Bayesian and frequentist statistics
- 100 - is small only when... - The implication appears to be the wrong way round. Small $d_{ij}$ could happen by chance - trivially, consider the constant graphon / Erdos Renyi graph - $d_{ij}$ would be zero everywhere

Typographical and grammatical errors:
- 25 - size of [the] graph
- 36 - 'we call it' -> 'is called a'
- References 12 and 13 D. Lawrence -> Neil D. Lawrence
- Reference 15 *Estimation...*block structures
Summary: A mostly clear paper on frequentist estimation of graphons demonstrating consistency of their estimator. Some missing references and remarks I would have expected.

Submitted by Assigned_Reviewer_5

The authors describe a new method for predicting a family of random graph generators (graphons which are completely described by a function w:[0,1]^2 -> [0,1]) from a series of observed graphs generated from the random graph generator. They develop a method that approximates the function w using step functions for the case where the true function w is piecewise Lipschitz.

The paper takes a new approach to a previous method in Lloyd et al, avoiding the need to place a prior distribution over w. Instead their approach is to apply a clustering method to group similar vertices across all the observed graphs and use averages in these blocks to approximate w by a step function.

They compare their new method against other methods for predicting the random graph generator and show significant improvement over a range of different scenarios.

Quality
----------------
This paper has many spelling mistakes, typos and layout problems, including the title which I think should be perspective and not perpective. I suggest that the authors give this paper a couple of proof reads to eliminate the errors. The ones I found whilst reading this paper are

In the paper:
- The title
- Sentence just above eqn 5
- bottom of pg 3 has the title of a subsection with no text below it
- The last sentence before section 3
- First sentence at the top of pg 6 references an appendix I think the authors mean the supplementary material
- Typo in eqn 10 w(u_i_y,j_v) I think should be w(u_i_y,u_j_v)
- Theorem 3 uses S\in\theta(n) and then subsequently the text refers to S_n. I think the authors mean S_n here since I believe it is meant to be a subset of S = \{1,....,n\} \ \{i,j\}.
- Second sentence of sect 4.1.2 send -> end

In the supplementary material
- ?? in theorem 2
- Lemma 3 mentions a proposition 8 and there is none
- eqn 5 has a typo (see note about eqn 10 in list above)
- The statement of Lemma 2 is grammatically incorrect
- The statement of Lemma 3 is grammatically incorrect
- Theorem 3 again states S but then uses S_n
- p10 bottom half mentions a lemma 8 and there is none (I think they mean lemma 4)
- p12 second eqn up from bottom d_ij is missing a superscript c
- p12 last equation the third f_ij does not need a sup


Clarity
--------------------------
In general, typos and spelling mistakes aside, this paper is coherent and clear. There were, however, a few points that could be made clearer

In the Paper:
- The use of S and S_n was not clear and I found this confusing, I think that this is possibly a typo (see above)
- The first experiment 4.1.1 compares their method to others for an arbitrary graphon. To make the comparison fair the authors use graphs of size n for the other methods and two graphs of size n/2 for their method. The second plot showing how their method improves with increasing numbers of observed graphs then seems to use graphs of size n. This could be made clearer.
- The discussion of graphs with missing links talks about applying a random binary matrix M to the observed graphs to eliminate some of the links. It was unclear if the M remained constant across all the observed graphs G_1...G_2T and then was changed for each set of observed graphs or if a new M was used for each of the G_1 .. G_2T. If it was a the first way round this would lead to a consistent bias in the observed graphs (since vertices would consistently have missing links throughout). I think the latter approach of randomising M for each graph would be more natural approach.

In the supplementary material:
- I felt that since lemma 1 - 4 were proved before theorem 3 it would make sense to place lemma 5 here also.

Originality
-------------------------------
The paper seemed original but I have had limited exposure to this area of research and so can't be certain.

Significance
---------------------------------
The paper seemed to be significant, but again I have had a limited exposure to this area of research so I cannot be certain.
Summary: This paper develops an interesting technique for predicting a family of random graph generators from a series of observed graphs. It improves substantially on earlier methods. Although it presents good results the quality of the paper is lacking, it has many spelling mistakes and I feel needs a thorough proofreading before final submission if it is to be accepted.

Submitted by Assigned_Reviewer_6

This paper presents a method to estimate the underlying graphon (the limiting object of an exchangeable random graph) based on the stochastic block model. The presented estimator is shown to be unbiased and consistent. Synthetic examples showed the effectiveness of their algorithm.

Overall this was a good paper with clear descriptions of the estimator, its theoretical properties and the experiments. However, do we ever observe multiple independent graphs from the same graphon in the real world? In most settings it is assumed that only a single observation of a graph exists and subsequently most models for graphs (including ones cited in this paper) only require a single observation of the graph. This paper addressed this in one of the experiments by sub-sampling a graph, however, this is slightly unsatisfying.
Summary: A nice paper that contributes frequentist nonparametric methods for network analysis.
Author Feedback

Author rebuttal: We thank all the reviewers for their valuable comments.

The contributions of this paper are: (1) we develop a frequentist approach to the graphon estimation problem, which to our knowledge is the first such result; (2) we prove consistency of our estimator, which is something that has not been done for any of the existing Bayesian approaches to this problem.

Reply to Reviewer 4

A representative element w of a graphon is unique only up to measure preserving transformations. Our estimator doesn't conflict with this idea because: (1) the estimator itself is defined up to permutation of the nodes (hence there exists a measure-preserving transformation of the graphon that generates the same graph); (2) the position of the u_i’s do not matter --- what matters is the size of the cluster and how the nodes are clustered. Note, for instance, that notion of error we use to define consistency doesn't depend on the u's, but on the values of w on the u's.

Reply to Reviewer 5

Lloyd et al address the same problem. The main differences are (1) in Lloyd’s paper, a Bayesian approach was used and a prior of the graphon was assumed, whereas our paper is a frequentist approach; (2) most importantly, there was no theoretical analysis of Lloyd’s paper, whereas we are able to derive the consistency guarantee of our approach. Note that the consistency proof is for the entire estimator, including both the clustering step and the graphon estimation step.

Reply to Reviewer 6

We do have more results for 1 sample and we can include them. Please suggest if this is important, and what should be removed to make space. We also do have a method for 1 sample that is consistent, but we decide to save that for a separate paper.

Having said that, it is arguable whether or not the 1 sample or the 2 sample situation is more relevant to science and industry at large. In the corporate situations we have been dealing in the past, the population is always well defined, possibly increasing over time and the temporal measurements are available on the same population. Thus the 2 or multiple sample situation is relevant. It is true, however, that a number of publicly available datasets on large and small networks are released with only 1 sample. We would argue that methods for 1 sample and 2 samples are both useful in practice.

In terms of design of new experiments, our 2/multiple sample method allows us to address an additional question which we find interesting: what are the advantages of collecting 1 large sample vs 2 or more smaller samples on the same sub-network? The answer, in brief, is that replicated networks help estimate a population graphon more precisely. While this is not surprising, it is a good reminder to people who focus collecting and releasing single samples of large networks, asserting that a single larger sample is better.